# Influence of Hydrometeorological Hazards and Sea Coast Morphodynamics onto Development of the *Cephalanthero rubrae-Fagetum* (Wolin Island, the Southern Baltic Sea)

Jacek Tylkowski[1], Marcin Winowski[1], Marcin Hojan[2], Paweł Czyryca[1], Mariusz Samołyk[1]

[1] Institute of Geoecology and Geoinformation, Faculty of Geographical and Geological Sciences, Adam Mickiewicz University, Krygowski 10, 61-680 Poznan, Poland

[2] Institute of Geography, Department of Landscape Geography, Kazimierz Wielki University, Kościeleckich Square 8, 85-033 Bydgoszcz, Poland

*Correspondence to*: Jacek Tylkowski (jatyl@amu.edu.pl)

**Abstract**: Climate changes, sea transgression and sea coast erosion observed today cause dynamic changes in coastal ecosystems. In the elaboration, cause and effect interrelations between abiotic hazards (hydrometeorological conditions and sea coast morphodynamics) and biotic (*Cephalanthero rubrae-Fagetum* phytocoenosis) components of natural environment have been defined. An up-to-date phytosociological analysis of a very valuable *Cephalanthero rubrae-Fagetum* site on cliff tableland was conducted in the context of hitherto temporal variability of climatic conditions and the rate of cliff coast recession. Also, the development prognosis of the researched site in the 21st century is provided, with respect to the expected climate changes and cliff's morphodynamics. The conducted research actions revealed the influence of global hazards (e.g., climate changes, sea transgression and sea coast erosion) onto changes in natural environment on regional scale (with the example of the site of *Cephalanthero rubrae-Fagetum* on cliff coast of Wolin Island in Poland). It has been established that in the 21st century, a relatively larger hazard to the functioning of the researched site are climate changes (i.e. mostly changes in thermal and precipitation conditions) not the sea coast erosion.

**Key words:** hydrometeorological hazards, climate change, sea coast morphodynamics, coastal vegetation

## 1 Introduction

Contemporary researches confirm dynamic climate changes, which are evidenced mainly in rise of temperatures (Sillmann et al., 2013). The result of thermal climate changes is the rise of sea level by approximately 2 mm yr$^{-1}$ (Church et al., 2013). The temporal variability of hydrometeorological conditions is decisive for the sea coast erosion dynamics and causes changes in coastal phytocoenoses (Strandmark et al., 2015). A particular role in this respect is reserved for extreme hydrometeorological events (Tylkowski and Hojan, 2018). Intensification of geomorphological processes, in the majority of cases, results in degradation of coastal vegetation sites (Feagin et al., 2005). Exceptionally rapid and intensive changes of natural environment are present in poorly resistant to erosion, moraine cliff coasts of the Baltic Sea (Kostrzewski et al., 2015).

That is why empirical researches on the influence of abiotic conditions onto determination of current state, threats and
development perspectives of all coastal phytocoenoses are particularly important.

32         Unique in the world are the sites of coastal thermophilous orchid beech wood, *Cephalanthero rubrae-Fagetum (Cr-*

*F)*, which are found only in Poland, on cliff coast of Wolin island, in Wolin National Park. *Cr-F* grows on specific soils and
is a peculiar type of beech wood, recognised as separate regional association (Matuszkiewicz, 2001, 2014). The uniqueness of
this plant community stems from endemic and specific character of habitat formation. *Cr-F* occurs on the top of the cliff (the
so-called 'cliff top') and on cliff tableland, where unique, rich in calcium carbonate soils in the form of cliff naspa were formed.
Naspa's accumulation level consists in interbeddings of fine-grain sand and dust drifted by wind from eroded cliff slopes, and
rich in humus, dark-grey organic accumulation laminas (mainly leaves of *Fagus sylvatica*). The cliff naspa is a soil with
reaction close to neutral, rich in calcium carbonate and characterised by high porosity and efficient humification of organic
remains. That is why naspa is a fertile soil. Naspa is deposited on the fossil podzolic soil. Naspa has the following sequence
of soil levels: A0 litter level; A1I accumulation level of sand and organic matter layers; A1 (fos) accumulation level of fossil
podzolic soil; A2 (fos) eluvial level of fossil podsolic soil; B (fos) iluvial level of fossil podzolic soil; C (fos) parent rock of
fossil podzolic soil (Prusinkiewicz, 1971). Therefore, the prerequisite for the development of this phytocoenosis is its non-
episodic, aeolian supply of mineral material from clayey and sandy cliff slopes. Moreover, the dynamics of cliff coast recession
may not be too extensive, as spatial reach of *Cr-F*, counted from cliff top, is 150 m at maximum (Piotrowska, 1993). The
average rate of aeolian deposition in the *Cr-F* habitat was 3-5 mm y$^{-1}$, and the maximum point value was 8-10 mm y$^{-1}$ (2000-

47 2019).

48         *Cephalanthera rubra* and *Epipactis atrorubens* are indicator species for *Cr-F* (Matuszkiewicz, 2020). Both species

found in the 6 studied *Cr-F* habitats, but *Cephalanthera rubra* was the dominant one. Non-indicator species, e.g.
*Cephalanthera damasonium* and *Epipactis helleborine*, have been found in *Cr-F* habitats too. The researches on *Cr-F*
conducted up to now (among others, Czubiński and Urbański, 1951; Piotrowska, 1955, 1993) were concentrated mainly on
qualitative floristic and phytosociological analysis. On the other hand, the main aim of this elaboration was the up-to-date
evaluation of the plant richness and floristic composition of *Cr-F*, and possible growth of this exceptional association, in the
context of climate changes and morphodynamics of cliff coast expected to take place in this century.
**2 Study Area and Methods**

56         The section of cliff coast, in which *Cr-F* occurs, was developed as a result of undercutting Wolin end moraine by the

transgressing Baltic Sea. Ultimately, orchid beech wood sites have been developed on hinterland of moraine cliffs. Moraine
cliffs at *Cr-F* sites are characterised by high morphological (height of 20–95 m, dominant NW exposition, inclinations op to
1° on cliff top, and up to 88° on clayey slopes) and lithological (sandy sections, clayey or mixed — sandy and clayey)
differentiation. The analysed section of cliff coast with the length of merely 3 km features various morphodynamic states
(erosion or stagnation). The researched site type is rich in species characteristic for, both, forest and non-forest phytocoenoses.
Forest species, characteristic for *Fagetalia* and *Querco-Fagetea* as well as meadow species with *Molinio-Arrhenatheretea*
occur in large numbers (Piotrowska, 1993). The high flow of light to the ground from the sea direction favours the occurrence
on the top cliff of many heliophilous species, characteristic for meadows and psammophilous short-grass swards. Gramineous
species prevail in the herb layer, among others: *Brachypodium sylvatica, Dactylis glomerata, Poa Nemoralis*. The most
valuable are orchid species, *Cephalanthera damasonium, Cephalanthera rubrae, Epipaptis atrorubens*, which prefer fertile
soils with reaction close to neutral (Piotrowska, 2003). There are, however, no of the numerous species characteristic for
*Fagetalia sylvaticae* order (*Actaea spicata, Daphne mezereum, Lathyrus vernus, Mercurialis perennis*) and *Querco-Fagetea*
class (*Aegopodium podagraria, Campanula trachelium, Corylus avellana*) that feature considerable share in all other
*Cephalanthero-Fagenion* forests, which evidences the distinction and uniqueness of the *Cr-F* association (Matuszkiewicz,
2001). The source of Latin names of plant species and plant communities are the publications Jackowiak et al. (2007) and
Matuszkiewicz (2020).
The current reach and floristic composition of *Cr-F* has been determined on the basis of a few phytosociological
mapping conducted on 6 study sites over 2018 and 2019 vegetative seasons. All in all, 10 detailed phytosociological images
were taken with the use of Braun–Blanquet method, and *Cr-F* habitats reach chart on Wolin island was drafted (Fig. 1). An
assumption was adopted that *Cr-F* site reach is determined by soil conditions. The cliff naspa determines the occurrence of
*Cephalanthera rubra* and *Epipactis artorubens*, which are species regionally characteristic of *Cephalanthero rubrae-Fagetum*.
The site's reach limits are indicated on the basis of occurrence of *Cephalanthera rubra*.

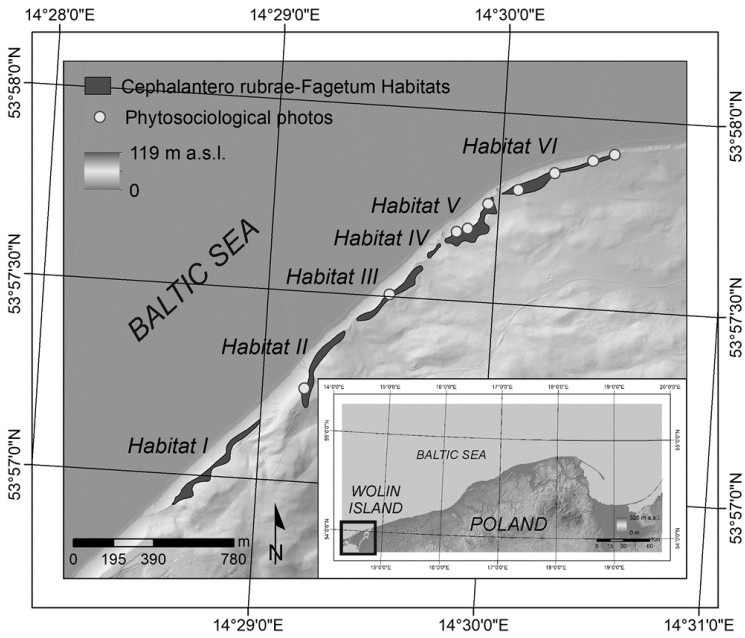


**Figure 1.** *Cr-F* habitats, localisation of phytosociological mapping on Wolin Island.

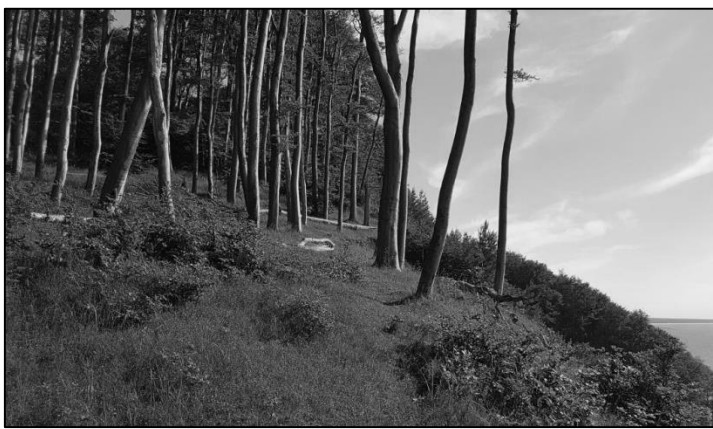


**Figure 2.** *Cr-F* habitat II on Wolin Island.
Detailed recognition of hydrometeorological conditions and the recession rate of the cliff top are vastly important for
the functioning of *Cr-F* habitats. Thermal and precipitation conditions determine, e.g. on water and heat resources and duration
of vegetative season. On the other hand, extreme storm surges may generate intensive cliff erosion and consequently reduce
the spatial extent of coastal plant communities. Therefore, unfavorable hydrometeorological conditions may limit the
development of the *Cr-F* habitats. For the purpose of defining long-term trend for thermal and precipitation conditions and sea
level, daily hydrometeorological data in the period of 1960–2019, collected in measurement station in Świnoujście, were used.
The data were provided by the Polish Institute of Meteorology and Water Management. The meteorological and
mareographical station in Świnoujście is located 15 km from the research area and provides homogeneous and complete series
of actual data.
In the elaboration, a number of especially useful climatic indicators were calculated and their values compared with
threshold values adequate for *Fagus sylvatica* given by Budeanu et al. (2016):
- De Martonne Aridity Index IA=P/(T+10), where *P* is the amount of the annual precipitation, *T* is the average annual
temperature (De Martonne, 1926); with optimal thresholds for beech wood in the range of 35–40 (Satmari, 2010); De
Martonne Aridity Index - classification by Tabari et al., (2014): IA<5 extremely arid, 5<IA<10 arid, 10<IA<20 semi-arid,
20<IA<24 mediterranean, 24<IA<28 semi-humid, 28<IA<35 humid, 35<IA<55 very humid, 55<IA extremely humid.
- Ellenberg Quotient EQ=Tw/Px1000, where *Tw* is the temperature of the warmest month of the year, *P* is the annual
precipitations (Ellenberg, 1988); with optimal threshold beneficial for beech growth of below 30 and its recession threshold
of above 40 (Stojanovic et al., 2013),
- Forestry Aridity Index FAI=100x($T_{VII-VIII}$/($P_{V-VII}$+$P_{VII-VIII}$)), where $T_{VII-VIII}$ is the average temperature of the months July and
August, $P_{V-VII}$ is the amount of precipitations during May-July and $P_{VII-VIII}$ is the amount of precipitations during July-August;
with climatic conditions favouring beeches of below 4.75 (Führer et al., 2011),
- Mayr Tetratherm: MT=($T_V$+$T_{VI}$+$T_{VII}$+$T_{VIII}$)/4, where $T_V$-$T_{VIII}$ represent the mean temperature for the May-August period
(Mayr, 1909); with optimal thermal conditions for beech wood of 13–18°C (Satmari, 2010).
The main zone of *Cr-F* occurrence is the cliff top, which changes its location as a result of, among others, mass
movements, water erosion and aeolian erosion. Thus, the cliff's morphodynamics is decisive for spatial reach of *Cr-F*. Annual
measurements of the recession rate of cliff top and evolution of slope forms have been conducted since 1984 on four orchid
beech wood sites (Fig. 1), (Kostrzewski et al., 2015; Winowski et al., 2019). Geomorphological changes in the cliff coast were
registered a few times over a year, based on geodetic measurements, geomorphological mapping, photographic documentation
collected with the use of photo-traps and drones.
**3 Results**
**3.1 Hydrometeorological Conditions and Hazards**
In the researched 60-year period (1960-2019), the mean annual air temperature reached 8.7°C, with statistically
significant rising trend of 0.3°C per 10 years (Fig. 3). A cooler period lasted until the end of 1980s. Since 1990s, a considerable
warming up may be observed, and especially warm period has been the decade of 2010s. The mean annual precipitation
reached 546.7 mm. Annual sum of precipitation has not shown statistically significant long-term trend (Fig. 3). However, for
the mean and maximum annual sea level, statistically significant rising trends in their values have been observed. The mean
sea level has been rising by 2 cm per 10 years, which correlates with the results of Church et al. (2013). On the other hand, the
dynamics in the maximum level rise is twice as high and amounts to 4 cm per 10 years (Fig. 3). Such positive long-term trends
evidence a rising threat of cliff coast abrasion in the future. The mean annual sea level in the period of 1960–2019 amounted
to 501 cm, but in the last 10 years it reached 508 cm.

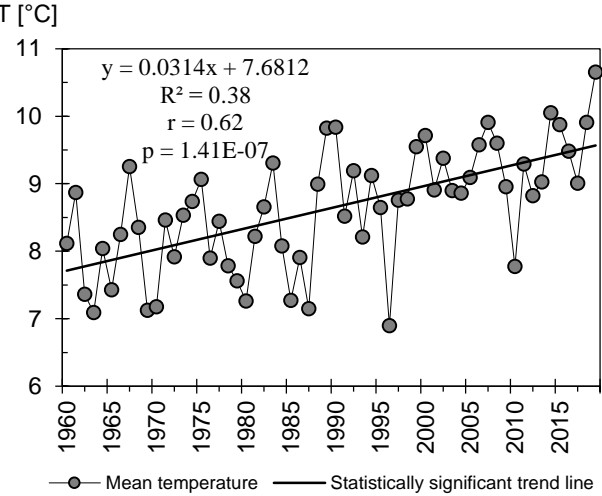
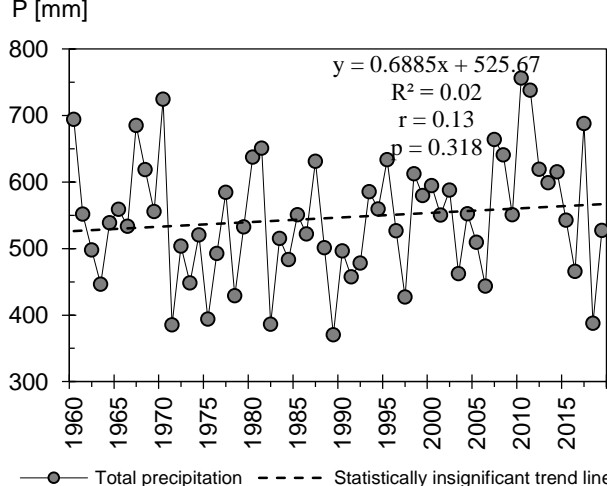


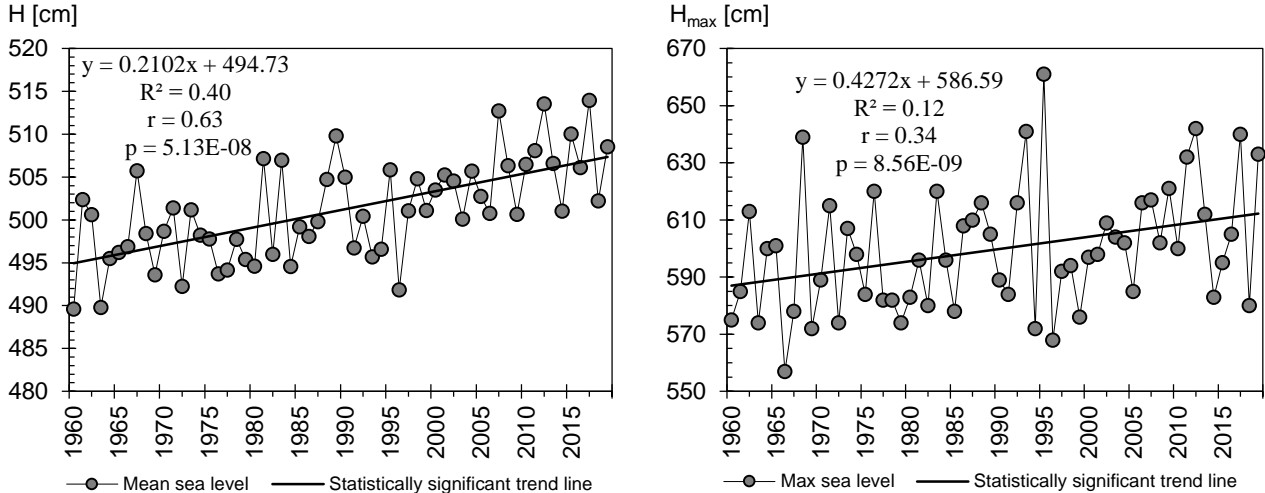

**Figure 3.** Long-term trends in hydrometeorological conditions: annual mean air temperature (T), annual total precipitation (P), annual mean sea level (H), annual maximum sea level ($H_{max}$), (Świnoujście, 1960–2019). *(Own study based on raw data from the Institute of Meteorology and Water Management in Warsaw).*

For recognition of thermal conditions of floral growth, a detailed analysis of thermal conditions trend may be presented with the data on vegetative season and heat resources. In Poland, the vegetative season starts, when the man daily air temperature exceeds 5°C. Heat resources in the vegetative season may be presented with the sum of effective temperatures, which are the sum of surpluses of the mean daily temperature exceeding 5°C (Tylkowski, 2015). The vegetative season in the research area lasts, on average, 228 days; it usually starts on March 30 and ends November 12. A statistically significant trend of extending the vegetative season by +3 days per 10 years has been proved (Fig. 4). The mean annual (1960–2019) sum of effective temperatures reached 1,817°C, and annual range of variability amounted to 1,500°C in 1967, and up to 2,254°C in 2018. The indicator of effective temperature sums featured for the researched area a positive trend of heat resource rise by 60°C per 10 years (Fig. 4), which is a favourable condition for the growth and expansion of stenothermal species. A regularity of a considerable heat resource rise has been confirmed, especially over the last 20 years. The dynamics of increasing the heat resources, especially in the 21st century is more noticeable than the increase in duration of the vegetative season.

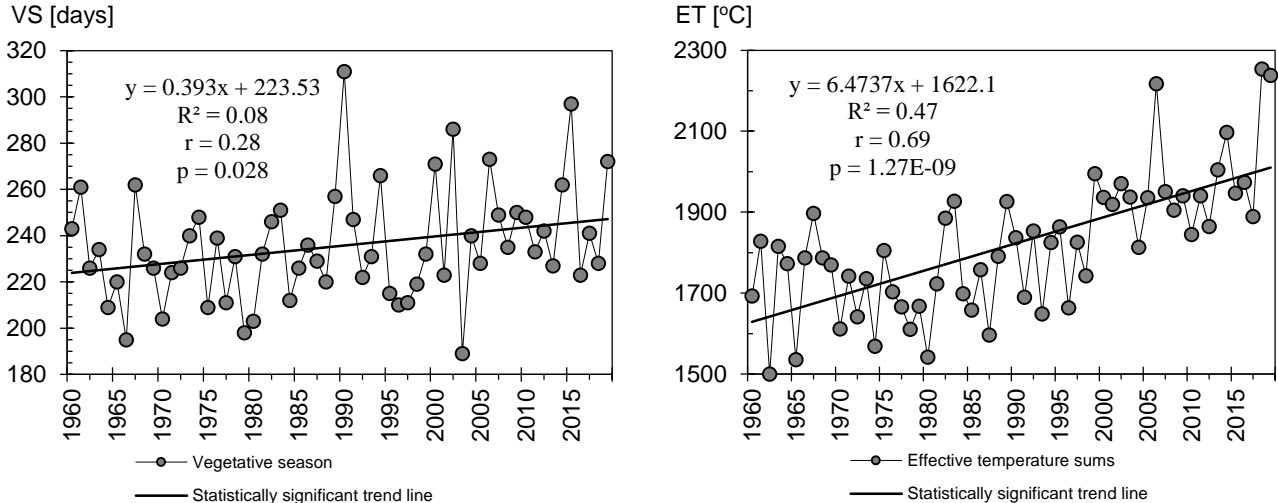


**Figure 4.** Long-term trends in the length of vegetative season (VS) and effective temperature sums (ET), (Świnoujście, 1960–2019). *(Own*
*study based on raw data from the Institute of Meteorology and Water Management in Warsaw).*

142       In the last 60 years, the AI, EQ and MT indicators confirm long-term trend of worsening climatic conditions for
*Cr-F* (Fig. 5). The AI and FAI indicators point to statistically insignificant (p>0.05) dropping trend, and the EQ indicator -
insignificant rising trend. The proven long-term regularities of these indicators suggest worsening thermal and precipitation
conditions for the researched forest phytocoenosis in subsequent years of the 21st century. Climatic indicators will probably
head towards the threshold values for sub-humid conditions (AI index), which will spur the decay of beech forest (EQ index).
Unfavourable thermal conditions will grow especially rapidly in the vegetative season (MT index), for which a statistically
significant rising trend (p>0.05) has been established with the value of 0.33°C per 10 years (Fig. 5). Taking into account this
trend's continuance in the future, it should be expected that within approximately 50 years, the thermal conditions for
occurrence of *Cr-F* will be too excessive, and as a result, its degeneration will advance. Analysis of agro-climatic indicators
(Fig. 5) pictured that during phytosociological mappings of *Cr-F* in 2018 and 2019, highly unfavourable climatic conditions
occurred for its functioning.

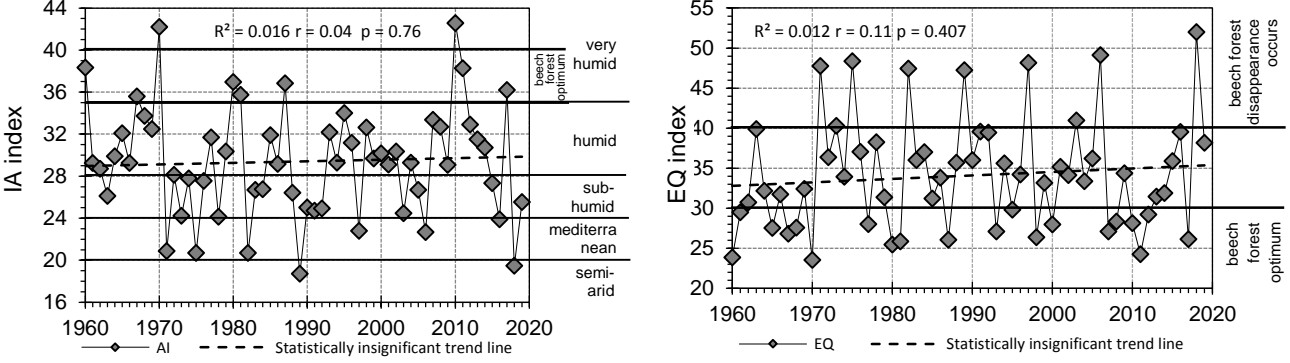


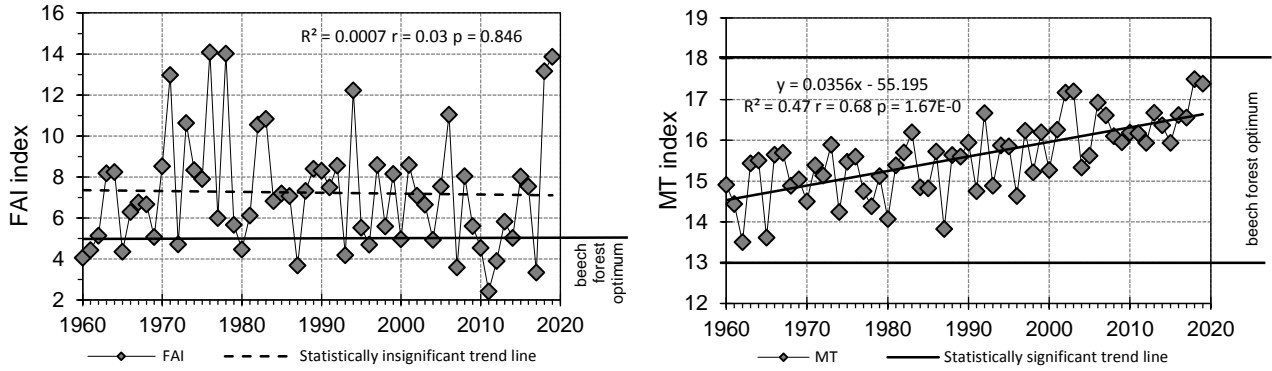


**Figure 5.** Long-term trends in climatic indicators: De Martonne Aridity Index (AI), Ellenberg Quotient (EQ), Forestry Aridity Index (FAI), Mayr Tetratherm Index (MT), (Świnoujście, 1960–2019). *(Own study based on raw data from the Institute of Meteorology and Water Management in Warsaw).*

### 3.2 Cliff Coast Morphodynamics Hazard

The mean annual rate of cliff top recession in 1984–2019 at *Cr-F* habitats II, III and V amounted to 0.24 m yr$^{-1}$. The lowest mean annual value of cliff recession was measured for site V (0.12 m yr$^{-1}$), where the cliff is built mainly of clayey sediments. The clayey sediments are characterised by relatively high resistance to degradation processes and the reaction time of cliff top to abrasion undercuttings is extended. A large number of storms is needed for the damages to reach the cliff top. On the other hand, the highest rate of cliff erosion has been established for site III (0.31 m yr$^{-1}$), where the cliff is built mainly of sandy material that is non-resistant to erosion. Sandy sediments are characterised by very low cohesion and are subject of rapid degradation. During stormy swellings, the sandy cliffs are undercut in a short time, which favours initiation of aeolian processes (deflation) and mass movements (sheddings, slidings). The processes cause the sediments to move across the entire slope profile, and thus the reaction of cliff top to abrasion undercutting is relatively short. An increased erosion dynamics has been observed also in site II (0.27 m yr$^{-1}$), on the cliff built of, both, clayey and sandy sediments. Its characteristic feature is the occurrence of underground water effluences, and high humidity of clayey sediments increases the susceptibility to landslide processes. The efficiency of the cliff springs is rather small <1 dm$^3$ min$^{-1}$. Landslide processes generate the highest cliff's transformations, contributing to movements of its top and cause reduction of *Cr-F* site area. In total, over the last 35 years, the researched cliffs recessed by an average of 7.32 m. The rate of recession of cliff top was spatially varied. The largest local and pinpoint movements were measured in the western part of site II (28.44 m) (Fig. 6). In this location, owing to high activity of landslide processes, the cliff top recessed with a high rate of 0.81 m yr$^{-1}$. In turn, the smallest local movements of cliff top were noted for eastern and western part of site V (0.30–0.42 m). In these locations, a very small rate of cliff top recession was connected with high resistance of clayey sediments to erosion processes and amounted to merely 0.01 m yr$^{-1}$.

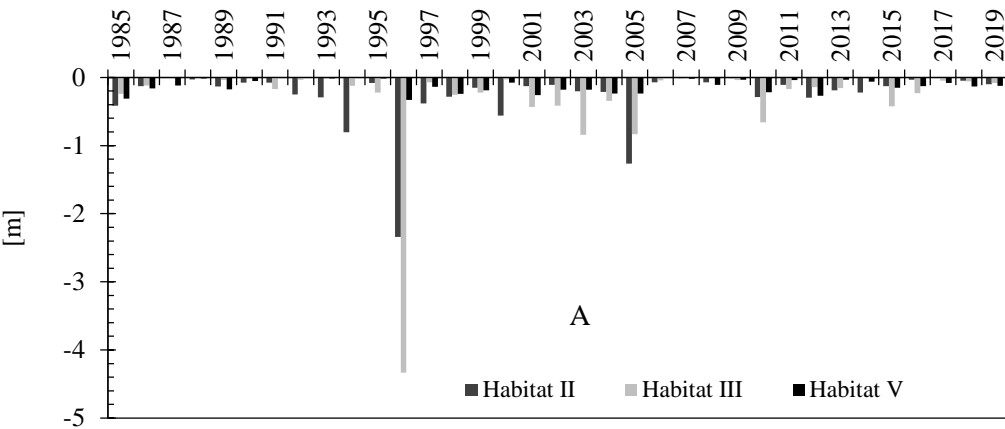

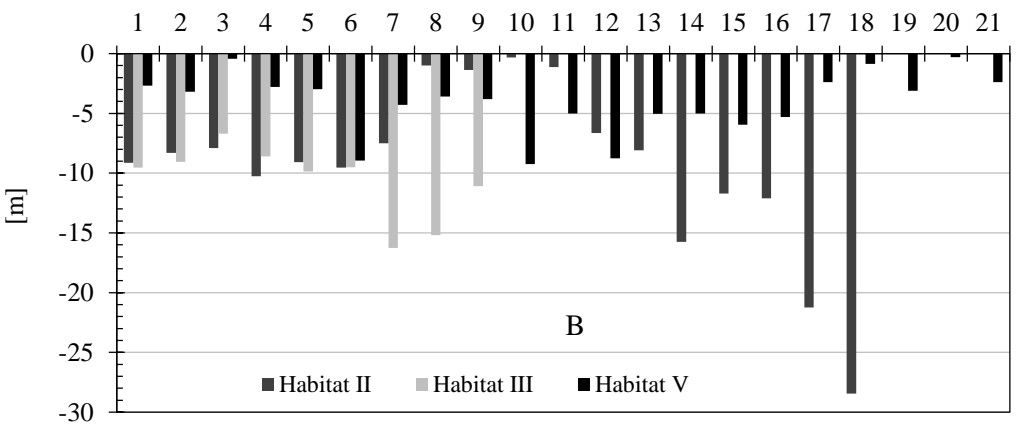

**Figure 6.** Location changes of cliff top at sites II, III and V of *Cr-F* in the period of 1985–2019: A – annual mean at sites, B – total multiannual in measurement points at sites. *(Own study based on own measurements and raw data from Kostrzewski et al. 2015, Winowski et al. 2019).*

A relatively lower sections of cliff coast (<30 m a.s.l.), which are primarily built of non-resistant to erosion sandy formations, do not favour the occurrence of the *Cr-F* phytocoenoses. In these sections of cliff coast, the deposition of sediments containing the calcium carbonate required by the orchid beech wood is relatively small (sandy sediments contain 4-5 times less calcium carbonate 2% than clay sediments) and an increased erosion (sandy sediments are much less resistant to erosion than clay sediments) of the coast results additionally in the reduction of habitat's area. A different situation is with the high cliff, with considerable share of clayey sediments. When aeolian processes occur, the dusty material, originating mainly in the clayey slope, rich in calcium carbonate, is accumulated on the cliff top and in cliff's hinterland, causing soil deacidification. This is the condition that particularly favours the development of *Cr-F* habitats (e.g., habitat V). Limited occurrence of the

orchid beech wood or its lack stems also from development cycles of the cliff coast. For the sandy and dusty material — that
is the components of the cliff naspa — to be supplied, a morphogenetic activity at the cliff's slope is required. Only then
material deflation from the cliff's slope and its subsequent aeolian deposition in the cliff's hinterland is possible. Thus, the
aeolian deposition is indispensable for the formation and development of the cliff naspa for inland. When the cliff coast, over
an extended period of time, is not subject to processes of maritime abrasion and slope erosion, then its slope is covered with
vegetation under of biocenotic succession. The vegetation considerably hinders, and even renders impossible the supply of
aeolian matter, and, in consequence, the formation of cliff naspa, which in a longer perspective spurs the decay of the *Cr-F*
phytocoenoses (e.g., habitat I). That is the occurrence of the active morphogenetic processes of small intensity is desirable
(e.g., at habitat V, mean annual rate of cliff top recession in the last 35 years amounted to 'as little as' 0.12 m yr$^{-1}$). The dynamics
of coast recession may not, however, be too intensive, and exceed the natural expansion of the cliff naspa and *Cephalanthero*
*rubrae-Fagetum* habitat for inland direction. Then, the decrease in habitat area is spurred (e.g., on habitat III, mean annual rate
of cliff top recession in the last 35 years has been considerable and amounted to 0.32 m yr$^{-1}$). Therefore, the optimal
morpholitodynamic conditions for the growth of *Cr-F* are found mainly on habitat V. Similar conditions are on habitats II and
IV. On the remaining habitats of the *Cr-F* phytocoenoses, the morpholitodynamic conditions are rather unfavourable - too
much (habitat III) or too little (habitat I) cliff erosion.

## 205 3.3 Reach and Floristic Composition of *Cr-F*

Currently, *Cr-F* grows along the northern cliffed coast of Wolin island, between Biała Góra and Grodno, in 6 isolated
sites with total area of merely 7.3 ha. The researched phytocoenosis occurs over a short, 3 km section of the coast, in the form
of narrow belt of approximately 100 m for inland, between cliff's edge and a complex of lowland acidophilous beech forest,
*Luzulo pilosae-Fagetum*.
The floral richness of *Cr-F* association consists in 113 species of vascular plants. They represent 2 divisions —
*Pteridophyta* and *Spermatophyta*. In *Pteridophyta* 4 species have been recorded: *Dryopteris carthusiana, Dryopteris filix-mas,*
*Polypodium vulgare* and *Pteridium aquilinum*. And, in *Spermatophyta* 3 classes have been confirmed: *Pinopsida* (2 species:
*Juniperus communis* and *Pinus sylvestris*), *Magnoliopsida* (23 orders, 29 families and 82 species) and *Liliopsida* (respectively
3, 6 and 27). The richest in species have been the families of: *Poaceae* (14 species), *Asteraceae* (13), *Fabaceae* (11) and
*Rosaceae* (6). *Orchidaceae* has been represented by 7 species: *Cephalanthera damasonium, Cephalanthera rubra,*
*Corallorhiza trifida, Epipactis atrorubens, Epipactis hellaborine, Neottia nidus-avis, Platanthera bifolia*. The researched site
is an example of a coexistence between forest species of fertile and acidic beech woods, acidophilic oak woods and forests,
and species of meadows and psammophilous swards. There have observed species from syntaxa: *Artemisietea vulgaris,*
*Festuco-Brometea, Molinio-Arrhenatheretea, Querco-Fagetea* and *Vaccinio-Piceetea*.

**Table 1.** Localisation and plant indicators of *Cr-F* habitats in 2019.

| Number of habitat | Habitat area [ha] | Habitat localisation Project Coordinate Reference System (CRS): WGS-84 EPSG code: 4326 | | | Plant indicators | | | |
|---|---|---|---|---|---|---|---|---|
| | | Western border | Geometric center point | Eastern border | Number of *Cephalanthera rubra* individuals | Population density of *Cephalanthera rubra* per ha | Number of vascular plants species | Number of orchid species |
| I | 1.6 | E 14.4773470193 N 53.9486589253 | E 14.4806801645 N 53.9506233460 | E 14.4834568531 N 53.9525988261 | 6 | 4 | 59 | 1 |
| II | 1.3 | E 14.4867629684 N 53.9532540446 | E 14.4874208216 N 53.9553690329 | E 14.4893115966 N 53.9566819942 | 57 | 44 | 97 | 4 |
| III | 1.1 | E 14.4901694844 N 53.9572079802 | E 14.4928896207 N 53.9585486797 | E 14.4946745712 N 53.9597487270 | 34 | 31 | 91 | 4 |
| IV | 0.1 | E 14.4951038446 N 53.9601527431 | E 14.4955996444 N 53.9604923130 | E 14.4959653287 N 53.9607642732 | 5 | 50 | 47 | 4 |
| V | 1.7 | E 14.4963451055 N 53.9608660790 | E 14.4985988815 N 53.9614999353 | E 14.4996322142 N 53.9629403030 | 51 | 30 | 73 | 6 |
| VI | 1.5 | E 14.5002867011 N 53.9631609678 | E 14.5046702332 N 53.9643211393 | E 14.5085083424 N 53.9651740858 | 22 | 15 | 78 | 5 |

Habitat I. The cliff slope is not subject to erosion processes, and for over 35 years it has been the so-called 'dead cliff'. Therefore, aeolian deposition on the cliff top is very limited and the *Cr-F* habitat decays. Soil profile and the presence of calcium carbonate in surface sediments confirm the presence of cliff naspa and morphodynamic activity of this cliff section in the past. On cliff top, there is only 6 *Cephalanthera rubra* specimens (Table 1), which are relics of a once well-developed habitat. There are no other orchid species found, though. The ground cover was poor (<5% coverage in the herb layer), and the confirmed species of *Luzula pilosa* and *Trientalis europaea* are the distinguishing species of the *Luzulo-Fagenion* beech forests.

Habitat II. In terms of phytosociology, this is a phytocoenosis of *Cr-F* typicum. The cliff wall is predisposed to aeolian processes as it is exposed and morphogenetically active. The ground cover is rich in species. The highest number (97) of vascular plants species was found in this habitat (Table 1). There is high concentration of *Cephalanthera rubra* (44 individuals per ha) and 4 orchid species have been found: *Cephalanthera damasonium, Cephalanthera rubra*, *Epipactis hellaborine*, *Epipactis atrorubens*. There are also numerous species of *Poaceae* family (among others, *Brachypodium sylvaticum*, *Calamagrostis arundinacea*, *Deschampsia flexuosa, Poa nemoralis*). Density of beech heads at this site is little (approximately 50%) and light conditions are favourable for the development of the ground cover (94% coverage in the herb layer), rich in species. A large portion (20%) of the site is covered by beech brushwood, which evidences an intensive renewal of forest.

Habitats III and IV. The plant indicators in Table 1 show that the habitats are moderately formed. At habitat III, there are intensive erosion processes taking place. Despite the aeolian deposition on the cliff top (40 m a.s.l.) is high, then due to a relatively high rate of cliff's recession (0.31 m yr$^{-1}$), the site's reach in this location decreases. The ground cover is well developed, and there are 4 species of *Orchidaceae*: *Cephalanthera rubra*, *Epipactis atrorubens*, *Epipactis hellaborine*, *Neottia nidus-avis*. They are, however, quite diffused and occur in a relatively narrow (*Cephalanthera rubra* density 31 individuals

per ha) strip along the cliff top (max 40 m). However, the habitat IV is a very small (0.1 ha), isolated area, where 5 individuals
of *Cephalanthera rubra* have been found.
Habitat V. The biggest patch of *Cr-F* typicum, developed the very good (Table 1). The cliff's wall is exposed, and high (35-
50 m a.s.l.) aeolian deposition on cliff top is visible. Aeolian material is visible on plants and the ground surface. The increment
of aeolian cover in the soil profile is about 4 mm $y^{-1}$ in 2000-2019. The ground cover is well developed (57% coverage in the
herb layer), rich in species (73), although in some areas their number drops due to poorer light conditions (high coverage of
forest canopy). There is a high abundance of *Cephalanthera rubra* (51), as well as other orchid species. This site is a strongly,
upon inland, encroaching part of the site. Species regionally characteristic for *Cr-F* have been found even up to 100 metres
from the cliff's edge. Even in this zone there were orchids, but their numbers were smaller than at the cliff. In total, 6 species
of *Orchidaceae* have been identified: *Cephalanthera damasonium*, *Cephalanthera rubra*, *Epipactis atrorubens*, *Epipactis*
*hellaborine*, *Neottia nidus-avis*, *Platanthera bifolia*.
Habitat VI. This habitat may also be considered a patch of *Cr-F* typicum (Table 1), but a smaller concentration of
*Cephalanthera rubra* (15 individuals per ha) has been confirmed there. The cliff is mostly clayey and low (25-30 m a.s.l.),
thus the intensity of aeolian deposition is relatively smaller (2 mm $y^{-1}$ in 2000-2019). The cliff tableland is flat. And the ground
cover covers up to 90 % of the phytocoenose area and is rich in species typical for orchid beech wood. There have been 5
species of orchid species from *Cephalanthero- Fagenion* confirmed: *Cephalanthera damasonium*, *Cephalanthera rubra*,
*Corallorhiza trifida, Epipactis atrorubens*, *Epipactis hellaborine*.
The most valuable orchid beech woods habitats are II, V and VI. Habitat V is the best developed patch of *Cr-F*, with
optimal habitat conditions: favourable morpholitodynamic conditions (abrasive coast but low rate of cliff's recession 0.12 m
$yr^{-1}$, higher share of clay sediments, rich in calcium carbonate 8-10%); favourable light conditions (relatively greater insolation
of the forest floor); ground cover of orchid beech wood, developing for inland for a dozen or so meters in some points). The
relatively poorest condition was confirmed for habitat I, which does not develop due to unfavorable morpholithodynamic
conditions (dead non-erosive cliff, stabilised with compact pine wood, no possibility of forming naspa).

## 4 Discussion

Current condition and future development of coastal phytocoenoses depends, primarily, on changes in climatic
conditions and morphodynamics of sea coasts. In the 21st century, in the Polish coastal zone of the Baltic Sea, the mean annual
air temperature may rise by 2–3°C, with concurrent rise in total precipitation by 0–10% during summer and 10–20% during
winter (Collins et al., 2013). Many research works indicate that in the last half-century, as a result of global warming (Sillmann
et al., 2013) the increase in activity of cyclones occurred, as well as the frequency of western winds in northern Europe (Pinto
et al., 2007) and over the Baltic Sea region (Sepp, 2009) increased. Another of the observed changes is the northward
displacement of trajectories of lows, which may cause advections of warm and humid air to northern Europe and decrease in
precipitation in central Europe (Bengtsson et al., 2006). The changes are connected with a varied location of the Icelandic Low

and the North Atlantic oscillation (NAO), (Omstedt et al., 2004). In the Baltic Sea catchment area, the warming will probably be higher than the mean global value, and the air temperature rise will, probably, be accompanied by higher precipitation, especially in winters. Also, the rise in frequency and duration of droughts (Orlowsky and Seneviratne, 2012) and heat-waves (Nikulin et al., 2011) is also expected. In the 21st century, the forecast climate changes will be accompanied by the rise in sea levels up to 1 m, and absolute rise of the Baltic Sea level is estimated to reach 80% of the mean rise of the world ocean level. For the south-west coasts of the Baltic Sea, the estimated rise in water level would be high, reaching approximately 60 cm (Grinsted, 2015). The executed hydrodynamic modelling iterations assume also the rise in frequency of stormy swellings for the entire Baltic Sea, in all seasons (Vousdoukas et al., 2016). Changes of the climate and hydrodynamic characteristics of seas will favour high frequency of extreme hydrometeorological events. In Poland, for the Baltic coasts, over the recent half-century, a rise in the frequency of extreme hydrometeorological events has been confirmed (Paprotny and Terefenko, 2017; Tylkowski and Hojan, 2018). Extremely high stormy swellings and precipitation intensify hydrological and geomorphological process, e.g., stormy floods or mass movements at cliff coasts. For the Polish coastal zone of the Baltic Sea, the occurrence of such unfavourable geomorphological results of extreme and above-average hydrometeorological events has been confirmed for, both, cliff and dune coasts (Florek et al., 2009; Furmańczyk et al., 2012; Hojan et al., 2018; Kostrzewski and Zwoliński, 1995; Tylkowski, 2017, 2018).

Climate changes in the 21st century will cause dynamic changes in the reach of forest phytocoenoses, including *Fagus sylvatica*. The forecast warming and gradual deterioration of water conditions in the coming 50 years will not influence considerably the changes in beech forest sites, yet. But from 2070 onwards, climatic conditions will be too warm and too dry for the growth of *Fagus sylvatica* and this species will start to withdraw from the area of researches (Falk and Winckelmann, 2013). The above forecast corresponds to the long-term trend of the agro-climatic indicators presented in the elaboration, especially with Mayr Tethraterm Index. According to the forecast variability of this indicator, in 50 years, climatic conditions will not be suitable for the development of the *Cr-F* habitat.

In the analysed period (1985-2019), the average annual rate of the cliff crown retraction on the examined sections amounted to 12 up to 31 cm and it was much lower than the values estimated (80-100 cm) by the mid-twentieth century by Subotowicz (1982) and Kostrzewski (1984). Whereas, the maximum annual point retraction of the cliff crown was almost 10 m. The average annual retraction rate of the Wolin cliffs is approximately 2-4 times lower than other monitored cliff coasts, e.g. in the vicinity of Ustka, Jastrzębia Góra or Gdynia (e.g., Florek et al. 2009; Łęczyński 1999). Although the Wolin cliffs are much higher and are not subjected to any protective measures, the relatively lowest rate of their retraction results primarily from specific hydrogeological conditions. For example, contrary to the cliff coast in Jastrzębia Góra (Uścinowicz et al. 2017) on the island of Wolin, underground waters practically do not play any role in erosion processes and shore degradation.

Species composition of association's phytocoenoses has not changed extensively over the last half-century (Piotrowska, 1993; Prusinkiewicz, 1971), which confirms its relative stability; however, some *Orchidaceae* habitats of do not keep up with the rate of the cliff's recession or they do not develop due to many years of cliff erosive stagnation. No specimens of *Malaxis monophyllos* were confirmed, which was occurring at the cliff's edge tens of years ago (Piotrowska, 1993). A vast

loss for the site is also the lack of current confirmation for the occurrence of *Listera ovata*. Also, it has been confirmed that the number of *Lonicera xylosteum* decreased — a species important for the orchid beech wood. In past elaborations, the indicatory species of *Cephalantero rubra* featured a larger reach in the area of Wolin National Park, e.g., in forest divisions of Międzyzdroje 16 and Wisełka 2. Currently, no specimens of *Cephalantero rubra* have been found on those sites, which is the confirmation for the decreasing reach of this species in Wolin National Park.

## 5 Conclusions

The analysis of *Cr-F* habitats indicated its small total area of merely 7.3 ha. This valuable site consists of 6 isolated, single sites with an area of 1.7 ha to just 0.1 ha. Discontinuity of the site stems from many natural factors — mainly due to the spatial variability of the cliff's morpholitodynamics. Phytosociological studies evidenced relatively good condition of *Cr-F* in majority of sites.

The analysis of temporal variability of hydrometeorological conditions, duration of the vegetative season and heat resources (1960–2019), as well as cliff coast morphodynamics (1985–2019) has indicated, up to now, rather favourable conditions for the growth of *Cr-F* site. A statistically significant trends of the increase in mean annual air temperature, sea level, duration of the vegetative season and heat resources have been verified. Analysis of climatic indicators AI, EQ and FAI in the last 60 years have not evidenced a trend of unfavourable climatic conditions clustering, and the occurrence of unfavourable thermal and precipitation conditions was of random character. Only the analysis of MT indicator pointed to an alarming and statistically significant rise in its value. It must be stressed that as of now, the regularities in long-term changes of AI, EQ indicators are unfavourable. Climatic conditions at the end of the 21st century may be too warm for *Fagetum* type forests, which — concurrently with uncertainty of precipitation efficiency and their time distribution — will intensify evapotranspiration and draught. It seems that climatic conditions of the southern Baltic Sea are heading for change in the 21st century from humid to subhumid, and in an even longer perspective — to mediterranean (IA index). Therefore, it is possible that access to water will be limited, and may influence a drastic change in the conditions of *Cr-F*.

As a result of global warming, the sea level rises, and in the future, this may be the cause of an intensified coastal erosion. Current cliff erosion rate is 0.3 m yr$^{-1}$. Thus, in the coming decade, the morphodynamic processes should not cause sudden degradation in the reach of *Cr-F* site. In a longer perspective, the dynamics definition of these processes is very difficult without precise recognition of submarine slope configuration and functioning of the circulatory cell system. Erosion process of the cliff coast are taking place over various time and spatial scales, and the highest erosion intensity is featured during extreme events that cannot be predicted. But, taking into account the increasing frequency of the maximum level of the Baltic Sea and stormy swellings, the erosion intensification of the sea coast may be expected. The development of *Cr-F* site is highly conditioned by the presence of cliff naspa and its formation due to aeolian processes. The cliff's erosive activity is a favourable condition for the development of the analysed site only to a certain degree. High activity of morphodynamic processes influences the high rate of cliff top recession, and this, in turn, contributes to the decay of *Cr-F* site area. On the other hand,

the limited influence of morphogenetic process favours the cliff's stabilization and sprouting of vegetation, and thus the *Cr-F* site does not develop. Therefore, the optimal condition for the development of *Cr-F* is the balanced cliff's dynamics. This notion is, however, difficult to be defined quantitatively due to high morpholitological diversity of cliffs. The simplest assumption is that the optimal condition for the growth of the orchid beech wood is the case, in which the cliff top recesses with a small, but stable rate of up to, approximately, 0.15 m yr$^{-1}$.

Future existence of *Cr-F* depends, primarily, on climatic conditions, and, to a lesser extent, on erosive process on cliff coast. Taking into account that *Cr-F* sites are found in the strict nature reserve of Wolin National Park, there is no need to introduce special protection measures. A favourable condition is the lack of cliff coast protection against erosive processes. Full limitation of cliff's erosion would result in lack of cliff naspa formation. As evidenced by multiannual field researches that have been conducted until now, more favourable conditions for the development of *Cr-F* are found in the cliff coast zone in erosion phase, and not stagnation, as the benefits stemming from aeolian accumulation and formation of cliff's naspa outweigh the losses in coastline due to cliff top recession.

**Data availability.** Data in this paper can be made available for scientific use upon request to the authors.

**Author contributions.** JT designed the research with participation of all the authors. JT and MW compiled data and conducted hydrometeorological and sea coast morphodynamics analyses. PC compile data and conducted phytosociological analysis. All other authors contributed with data or conducted a small part of data compilation or analysis. JT drafted the paper with participation from MH and comments from all authors.

**Competing interests.** The authors declare no competing interests.

**Acknowledgements.** The authors would like to thank the Polish Institute of Meteorology and Water Management in Warsaw for the provided hydrometeorological data. We would also like to thank the management of Wolin National Park, Marek Dylawerski and Stanislaw Felisiak, for their consent and assistance in scientific research. We also thank Natura company, especially Wojciech Zyska, for his help in drafting this elaboration.

**Financial support.** The research was supported mainly by the Forest Fund, within the scope of funding admitted by the Directorate General of State Forests National Forest Holding for Wolin National Park (agreement No. EZ.0290.1.21.2019 of 22 July 2019).

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
