# Peer review of "Influence of Hydrometeorological Hazards and Sea Coast Morphodynamics onto Development of the *Cephalanthero rubrae Fagetum* (Wolin Island, the Southern Baltic Sea)"

_Natural Hazards and Earth System Sciences, 2020_

## Short Comment (SC1) · 11 Jul 2020

Using data for Figures 2-5 without providing the owner of the data is an abuse of copyright. The authors of the article are too young to be able to conduct their own hydrometeorological, vegetative and morphodynamic studies in the years indicated.

[Figure]

2020-160, 2020.

---

## Author Comment (AC1) · 13 Jul 2020

Dear Mr. Welsh - The data in article is not abuse of copyright. - The study includes only raw, publicly available meteorological data and sea levels, which were obtained from the Institute of Meteorology and Water Management in Warsaw - data for ÅŽwinoujś-cie (1960-2019). In verses 85-87 the source of this data is given "For the purpose of defining long-term trend, daily hydrometeorological data in the period of 86 1960–2019,

collected in measurement station in Swinoujscie, were used. The data were provided by the Polish Institute of Meteorology and Water Management". The acknowledgments (verses 322-323) also include the source of hydrometeorological data "Acknowledgments. The authors would like to thank the Polish Institute of Meteorology and Water Management in Warsaw for the provided hydrometeorological data." - Only raw cliff morphodynamics data were used (Kostrzewski et al. 2015, Winowski et al.). The authors (Tylkowski, Winowski, Samołyk) are co-authors of these publications and have performed field measurements of the cliff recession rate since 2000. The sources of these data are provided in verses 322-323 "Annual measurements of the cliff-top recession rate and the evolution of slope forms have been conducted since 1984. At four beech sites (Fig. 1), (Kostrzewski et al., 2015; Winowski et al., 2019)." - All the figures in the manuscript are made by us and are original. Figures have not yet been published. Analyzes and figures were made only on the basis of raw data. - The authors with the editors of the journal also include data sources in the figures titles, e.g.Figure 2. Long-term trends in hydrometeorological conditions: annual average air temperature (T), annual total rainfall (P), annual average sea level (H), annual maximum sea level (Hmax), (ÅŽwinoujście, 1960–2019). (Own study based on raw data of the Institute of Meteorology and Water Management in Warsaw). Yours sincerely Jacek Tylkowski and co-authors of the article

---

## Short Comment (SC2) · 14 Jul 2020

Dear Mr. Tilkowsky and co-authors, It is glorious that thanks have been placed for the data at the end of the paper and in the text partly quoted where the data is from. However, scientific ethics, and first of all writing practice, requires that this information should be put there where they are used, presented, etc. In your case, in the captions. What we can see currently in the captions absolutely does not point to public data

and other authors but only to your own data! And this is this matter. This is obviously very confusing when it comes to assessing your contribution to science. I know that respecting other people's and public data for the young generation is not important nowadays. This can be seen in many contemporary papers but in papers of younger researchers only. Personally, I would advise you to follow good patterns developed over the years in the world of science. Many such cases of unjustified or undisclosed use of someone's data, unfortunately, ended in the courts. You may be violating the rule of law in your country, but if you publish in an internationally recognized journal, you should comply with international law. BTW: Figure2 does not contain the information you provide in your explanations. Good luck!

---

## Author Comment (AC2) · 14 Jul 2020

Dear Mr. Wolf Thank you for commenting on the article. Information about the data source will be placed in the titles of the figures. Yours sincerely Jacek Tylkowski and co-authors

2020-160, 2020.

---

## Author Comment (AC3) · 14 Jul 2020

Dear Mr. Welsh I am very sorry for your misspelled name in the AC2 reply. I had internet problems and I corrected the text incorrectly. Once again, I'm very sorry. Yours sincerely Jacek Tylkowski

2020-160, 2020.

---

## Referee Comment (RC1) · Tomasz Wolski (Referee) · 30 Jul 2020

General Comments

This is a valuable article and should be published in Natural Hazards and Earth System Sciences. The main aim of this elaboration was the up-to-date evaluation of the reach and floristic composition of Cr-F, and possible growth of this exceptional phytocoenosis, in the context of climate changes and morphodynamics of cliff coast. I believe that the

aim of the work has been achieved to a good degree. This article shows a good understanding of the authors of the geomorphological processes of the sea shore under the influence of extreme hydrological and meteorological factors. Additionally, the authors have extensive biological knowledge of the characteristics and formation of coastal ecosystems (Cephalanthero rubrae - Fagetum phytocoenosis). I consider it interesting and important that the authors emphasize the role of eolian processes in the formation of coastal phytocoenosis on the cliff. The conclusion of the article is accurate and relevantthe - optimal condition for the development of Cr-F is the balanced cliff's dynamics ( it means- cliff top retreats with a small, but stable rate of up to, approximately, 0.15 m/yr).

The authors of the article should only pay attention to the minor corrections I am presenting below.

Specific Comments

1. The authors of the study identified interesting climatic indicators (AI, EQ, FAI, MT). However, they were not well described. Please complete the formulas of these indicators. Please write how the value of a particular indicator influences the development (growth) of Fagus Silvatica.

2. Please explain this contradiction:

The authors wrote in the results (lines 181-184): "In the last 60 years, the AI, EQ and MT indicators confirm long-term trend of worsening climatic conditions for Cr-F (Fig. 4). The proven long-term regularities of these indicators suggest worsening thermal and precipitation conditions for the researched forest phytocoenosis in subsequent years of the 21st century."

The authors wrote in the conclusion (lines 286-288) "Analysis of climatic indicators AI, EQ and FAI in the last 60 years have not evidenced a trend of unfavorable climatic conditions clustering, and the occurrence of unfavourable thermal and precipitation

conditions was of random character."

3. Please add short words :

The authors wrote in Abstract (lines 19-21): "It has been established that in the 21st century, a relatively larger hazard to the functioning of the researched site are climate changes, not the sea coast erosion".

Please complete the conclusion - climatic changes, ie changes in thermal and precipitation conditions. Because increasing coastal erosion may also be a result of climate change. The same authors wrote in their conclusions (lines 295-296): "As a result of global warming, the sea level rises, and in the future, this may be the cause of an intensified coastal erosion.

---

## Author Comment (AC4) · 4 Aug 2020

Dear Mr. Wolski, Thank you very much for your review. We send answers to 3 questions.

1 The climate indicators AI, EQ, FAI, MT are described in the cited literature (lines 89-94). For Fagus Silvatica, the optimal and development-threatening threshold values of these indicators are also presented. The threshold values (1960-2019 climate data)

are also presented in Figure 4. Optimal weather conditions in individual years (AI, EQ, FAI, MT index values) favor the development of Fagus Silvatica. On the other hand, the values of these indicators outside this range are unfavorable for the development of the beech forest and may even cause the stand to disappear. It should be emphasized that episodic (in individual years) exceeding the optimal threshold values of the AI, EQ, FAI, MT indicators will not cause immediate beech degradation. Clustering of such events is more dangerous, and especially the long-term trend. Based on the analyzes of the AI, EQ, FAI, MT indicators in the period 1960-2019, no clustering of such unfavorable cases (several consecutive years) was found. The occurrence of unfavorable temperature and precipitation conditions (years) for the development of the beech forest was accidental. However, unfavorable climatic trends of these indicators were found, towards values unfavorable to the development of Fagus Silvatica. This situation will continue in the coming years in the 21st century. We give formulas for climate indicators (they will be placed in the article, in the methodological chapter):

De Martonne aridity index: $IA = P/(T+10)$ (De Martonne 1926), where P = the amount of the annual precipitation, T = average annual temperature. IA< 30 = silvosteppe, 30 < IA< 45 = climate favourable for the forest, with an optimal for beech in the range 35 - 40 (Satmari, 2010). De Martonne aridity index - classification Tabari et al., 2014: IA<5 extremely arid 5<IA<10 arid 10<IA<20 semi-arid 20<IA<24 mediterranean 24<IA<28 semi-humid 28<IA<35 humid 35<IA<55 very humid 55<IA extremely humid Ellenberg Quotient Index: $EQ=Tw/P\times1000$ (Ellenberg 1988) where Tw represents the temperature of the warmest month of the year, P = annual precipitations (Stojanović et al., 2013). Ellenberg (1988) has set a threshold of beech favourability for EQ values lower than 30, and at EQ values that are higher than 40, the beech disappearance occurs. Forestry Aridity Index (FAI): $FAI = 100 \times (TVII\text{-}VIII / (PV\text{-}VII + PVII\text{-}VIII)$ where TVII-VIII is the average temperature of the months July and August, PV-VII represents the amount of precipitations during May-July and PVII-VIII is the amount of precipitations during July-August (Führeret al. 2011). Mayr Tetratherm Index: $MT = (TV + TVI +TVII + TVIII)/4$ where tV – tVIII represent the mean temperature for the May-August period.

Tetratherm values indicate a climate restrictiveness of an area for some plant formations. Values between 13 and 18 degrees are optimal for the beech (Satmari, 2010).

2. There is no contradiction in the article between the results and the conclusions. The results in lines 181-184 and the conclusions in lines 286-288 are not inconsistency. The annual volatility of the AI, EQ and MT indicators from the 1960-2019 period shows an unfavorable trend of the deteriorating climatic conditions for Fagus Silvatica. However, no clustering (unfavorable conditions for consecutive years, e.g. 1965-1969 etc.) was found in this period. The occurrence of such years was accidental. When analyzing long-term regularities (trends 1960-2019), one should expect a deterioration of climatic conditions for Fagus Silvatica in the following years of the 21st century.

3. Of course, the threat to the development of the orchid beech on Wolin Island is both climate change (thermal and precipitation) and increased erosion of cliffs. However, climatic trends indicate that the adverse change in thermal and precipitation conditions will continue in the 21st century. The climatic factor will be dominant in the changes of the forest. Of course, climate change, its warming and the consequent rise in sea levels are likely to increase coastal erosion as well. However, there is currently no established trend in cliff erosion. The reversal of the cliff coast is mainly due to extreme hydrometeorological events that are episodic and random in nature. It is difficult to forecast at the moment about the dynamics of the recession of the cliff edge. Therefore, among the factors threatening the orchid beech, climate change plays a leading role in comparison with the erosion of the cliff coast.

Yours sincerely Jacek Tylkowski and co-authors of the article

---

## Referee Comment (RC2) · Tomasz Wolski (Referee) · 6 Aug 2020

I thank the authors for detailed and clear explanations of all my comments.

1 comment

It is good that the authors will place the formulas of climate indicators and their interpretations in the Methods chapter. This will enrich the work and make it easier for the

reader to understand the content.

2 comment

After the authors' explanations for my 2nd comment, I believe that there is no contradiction about which I wrote earlier.

3 comment

Please add a few words in bracket in the sentence in Abstract: (lines 19-21): "It has been established that in the 21st century, a relatively larger hazard to the functioning of the researched site are climate changes (ie mostly changes in thermal conditions and precipitation conditions) not the sea coast erosion" This will be clear to the reader.

After introducing these corrections to the final version of the work, I believe it is valuable and should be published in Natural Hazards and Earth System Sciences
* * *

---

## Short Comment (SC3) · 2 Sep 2020

The article undoubtedly deserves to be published. A very important research work. Both in the global context of trends in climate change and sea level rise. And also the interpretation and prediction of these phenomena for the southern Baltic Sea. The analyzes and interpretations performed are logical, methodically correct and well documented. They are based on own research and available source data. The reference

to literature is accurate and sufficient. Among other things, it is important that the reports of the Intergovernmental Panel on Climate Change (IPCC) were referred to. The conclusions are supported by the results. The importance of the content for research is significant. The advantage of the work is not only relying on own short analyzes of data sets. The authors also interpret commonly available archival data. Thanks to this, the interpretation is complete and credible. The authors analyze the data starting from 1960. These are the data of the Polish Institute of Meteorology and Water Management. The Polish Institute of Meteorology and Water Management operates in accordance with the World Meteorological Organization (WMO) guidelines. These are public data, obtained thanks to taxes from Polish citizens. Therefore, these data should be widely used in scientific research. The paper uses the commonly used phytosociological mapping method of a well-known Swiss phytosociologist and botanist Josias Braun–Blanquet. The so-called French-Swiss school or the Zurich-Montpellier school. The area of research has been correctly selected. It represents different morphological types of the coast and varied relative height and slope. The aspect also corresponds to the main course of the coastline SW-NE, WSW-ENE. The sites are characterized by different morphodynamics, different thickness of glacial sediments and aeolian deposits. Sea Erosion Rate results are available for most test sites since 1984. The presented test results are representative. Although the work concerns endemic phytocoenoses Cephalanthero rubrae-Fagetum, the research results are universal. They can be related to other areas of the South Baltic Sea. To the areas of cliffs with glacial genesis. A diverse team of authors allowed for a broad interdisciplinary approach to the presented issue. They were taken into account biotic and abiotic components of the environment. This is the right decision because in a given area, besides the well-known land-atmosphere interactions, the marine factor plays an important role. The authors proved, inter alia, there is a trend towards an increase in the average annual temperature of 0.3 degrees for every 10 years. They also showed that the trend for sea level rise in this area is 2 cm every 10 years. It is also important that the annual sums of atmospheric precipitation do not show statistically significant trends. Another

advantage of work is forecasting. The authors show which factors will have a greater impact on the range of Cephalanthero rubrae-Fagetum occurrence in the future.

Suggestions If the editor doesn't mind posting photos, it's worth adding photos of some endemic species from Cephalanthero rubrae-Fagetum. Question Winds from the W and NW directions dominate in the study area. Can it be assumed that the cliff recession rate in section VI (WSW-ENE) is lower than in sections I to V (SW-NE)? Or susceptibility to wind direction is less important than other factors?
* * *

---

## Short Comment (SC4) · 3 Sep 2020

The article presents a very interesting research on unique vegetation sites (Cephalantero rubrae – Fagetum, Cr-F) on Wolin Island in Poland. It should be emphasized that this is the only position of stenothermal coastal orchid beech wood in the world. Cr-F sites have been analysed on a two-year field study, which was related to the annual rate of the cliff top recession and evolution of slope forms (since 1984) and

hydrometeorological data from 1960-2019.

General comments:

The purpose of the study is clearly and precisely written but method chapter should be completed. There is no detailed information about the climate indicators used in the manuscript. Some parts of the text from the result chapter should be moved to the methods and discussion chapters (details below). Focus in discussing the specific results of this work. Conclusions chapter should be shortened, some part of the text should be moved to the discussion chapter.

Apart from the general comments I have listed some minor comments which the author might find useful to improve the manuscript.

Minor comments:

- page 2, line 42 –delete "on the other hand"

- page 4, line 94 – please remove the space between the number and °C

- page 6, line 159-160 – move the sentence to the discussion chapter

- page 7, line 168 – the first three sentences moved to the methods chapter

- page 7, line 174 – please change 1817°C to 1,817°C

- page 11, line 241-242 – please add references

- page 12, line 263 – please add a comma after the Tylkowski

- page 12, line 279-281 – in the conclusions chapter for the first time there is information about the range of occurrence of Cephalantero rubra in the forest divisions of Miedzyzdroje 16 and Wiselka 2. This sentence should be moved to the discussion chapter. Please add a citation.

- page 14, line 343 – add a comma after the name initial (Nauels, Y.,)

- page 15, line 369 – replace a comma to a dot (after the year 2015)

- Fig. 1 - you can add photo from the selected measurement site

- Fig. 2 – please present the precipitation as a bar graph

- Figs., 2,3,4 – please add regression equation, R2 and the level of statistical significance

- Fig. 4 – font size different than in the previous figures, please unify the description of the x and y axes throughout the article

- Fig. 5 – change order first place II, then III, V - as it is in the description

I recommended this valuable article for publishing in the Natural Hazards and Earth System Sciences after minor revision.

---

## Short Comment (SC5) · 3 Sep 2020

**General comments**

The manuscript deals with an interesting subject. In my opinion this is a valuable article and should be published in the Natural Hazards and Earth System Science. Apart from the comments in Interactive Discussion, I have a few minor remarks.

Line 100 Please add information on the statistical methods used.

Line 164 and 165 and 180 Please add R2 value, equation and statistical significance.

Line 189 The Authors wrote in line 189 "Agro-climate indicators" and in line 89 "climatic indicators". Do the Authors have the same indicators in mind? Please standardize terminology ?

Line 220 The Authors wrote: "In these sections of cliff coast, the deposition of sediments containing the calcium carbonate required by the orchid beech wood is relatively small...". Did the Authors examine the amount of the deposition size?

Line 270 These is no reference to the observed changes in the position of the cliff in the discussion. Please add discussion with other authors.

After introducing these corrections to the final version of the work, I believe it is valuable and should be published in Natural Hazards and Earth System Sciences.

---

## Short Comment (SC6) · 4 Sep 2020

Thank you for your comprehensive answer. As I mentioned at the beginning of my review, the article deserves to be published.

---

## Author Comment (AC5) · 4 Sep 2020

Dear Mr. Kolander Thank you very much for your review. Answering the questions: 1) A photo of the research area will be placed in the article. It is worth explaining that it is an endemic habitat of Cephalanthero rubrae-Fagetum, associated with the occurrence of unique soils - the cliffs naspa. Cephalanthero rubrae, also sold in other habitats. 2) The rate of the cliff retraction does not depend solely on the wind direction. The

most important source for cliff erosion are storm surges. Precipitation and snow thaws are of lesser importance for cliff erosion. Aeolian processes are relatively the least important, although they are most significant in uprooted tree hollows. Uprooted tree hollows causes significant point losses of the top cliff. Cause and effect relationships (hydrometeorological conditions - cliff edge erosion) are not directly proportional. Cliff erosion also depends on the current morphogenetical state of the individual sections, which may in erosion or stagnation phase. Therefore not always intense storm surges, extreme precipitation or strong winds cause extreme cliff erosion. There was no clear relationship between the cliff exposure towards the prevailing W-N winds and the dynamics of sea shore erosion. The current stage of the cliff section development and its lithology are greater importance. Moreover, storm surges from the NE direction of waves (from the open sea zone) have the greatest erosive importance. Waves coming from the west, from the Pommeranian Bay have less energy. Yours sincerely Jacek Tylkowski and co-authors
* * *

---

## Author Comment (AC6) · 4 Sep 2020

Dear Mrs Kijowska-Strugała Thanks a lot for your review. Your suggestions will be taken into account in the article publication process. Climate indicator formulas will be incorporated into working methods. Part of the text from the Conclusions chapter will also be moved to the Discussion chapter. Figures will be graphically corrected and statistical measures (R2, r, p and regression equation) will be added. Photos from

the research area will also be added and punctuation errors will be corrected. Yours sincerely Jacek Tylkowski and co-authors

---

## Author Comment (AC7) · 4 Sep 2020

Dear Mr. Kozłowski

Thank you very much for your review. Your comments will be taken into account in the publication process.

Line 100 Added information on the statistical methods used, incl. Mann-Kendall test,

regression equations, determination and correlation factors.

Line 164, 165, and 180 R2 value, regression equation, and statistical significance included in the figures.

Line 189 unified terminology as climatic indicators showing the impact of long-term weather conditions on the beech forest condition and development

Line 220 Detailed studies of aeolian deposition were conducted in the 2001-2004 years (e.g. Hojan M., 2009: Aeolian processes on the cliffs of Wolin Island. Quaestiones Geographicae 28/2: 39-46). The average annual rate of aeolian deposition on the cliff crown was almost 2 mm and the maximum (point) even 16 mm. The placed benchmarks showed an average aeolian deposition about 4-6 cm in the 2001 - 2020 period, with a maximum point increment of 10-12 cm.

Row 270 The discussion will compare the rate of cliff erosion on Wolin Island to other research sections in the South Baltic zone (e.g. cliffs in the vicinity of Ustka, JastrzĄŹbia Góra and Gdynia).

Yours sincerely Jacek Tylkowski and co-authors

---

## Referee Comment (RC3) · Anonymous Referee #2 · 28 Sep 2020

Comments placed directly in the manuscript pdf and in the evaluation table (text in word). Please share the review with both the editor and the authors.

Please also note the supplement to this comment:
https://nhess.copernicus.org/preprints/nhess-2020-160/nhess-2020-160-RC3-supplement.zip

---

## Author Comment (AC8) · 29 Sep 2020

Dear Reviewer Thank you for a comprehensive and insightful review. We agree with all the comments that mainly concern the geobotanical part. Terminological, linguistic, text and graphic comments will be included in the revision of the article. Best regards and thank you for your positive review.

Jacek Tylkowski and co-authors

---

## Author Comment (AC9) · 10 Oct 2020

Dear Reviewer Thank you for your positive review. All comments in the final version of the article will be taken into account. Yours sincerely Jacek Tylkowski and co-authors

---

## Author Comment (AC10) · 15 Oct 2020

All comments of the reviewer will be included in the article, e.g .: - title change: Influence of Hydrometeorological Hazards and Sea Coast Morphodynamics onto Development of the Cephalanthero rubrae-Fagetum (Wolin Island, the Southern Baltic Sea) - improving language terminology, e.g: stenothermal coastal - coastal thermophilous, phytocoenosis - plant community, site formation – habitat formation - comment line

41 - It will be added that Cephalanthera rubra and Epipactis atrorubens are indicator species for Cephalanthero rubrae-Fagetum. Both species found in the 6 studied habitats - Cephalanthera rubra was the dominant one. Non-indicator species, e.g. Cephalanthera damasonium and Epipactis helleborine, have been found in habitats - in the method section, the source of the Latin names of plant species and phytosociological units (Jackowiak et al. 2007 , Matuszkiewicz 2020). Jackowiak, J., Celka, Z., Chmiel, J., Latowski, K., Åżukowski, W.: Red list of vascular flora of Wielkopolska (Poland). Biodiversity Research and Conservation 5-8, 95-127, 2007. Matuszkiewicz, W.: Przewodnik do oznaczania zbiorowisk leśnych Polski [Guide to the determination of forest communities in Poland], PWN, Warszawa, 540, 2020) - habitat development (line 46-56) will only consider the formation of the present-day Cephalanthero rubrae-Fagetum habitat from the 18th century. The earlier context will not be described, as the former Cephalanthero rubrae-Fagetum habitats do not exist, they have been eroded - plant species in habitats will be listed alphabetically - line 77-80 will be transferred to line 35-39, with a detailed description of the naspa by Prusinkiewicz 1971 - replacing the uniform word to homogeneous word (line 88) - reordering in chapter 3 Results: 3.1. Hydrometeorological Conditions and Hazards. 3.2 Cliff Coast Morphodynamics Hazard 3.3 Reach and Floristic Composition of Cr-F - description of habitats: adding quantitative indicators (Cephalanthera rubra individuals recorded should be reported, population density indicator of Cephalanthera rubra, the number of species of vascular plants and orchids), writing the boundaries of habitat range (coordinates of the eastern and western boundaries of the habitat and its geometric center), a quantitative description of erosion rate in the last 35 years - linguistic inaccuracies will be corrected, e.g. the word movement in line 229 will be removed, - conclusions chapter: the term defragmented was used incorrectly instead of different places (line 272). Lines 275-282 quantitative changes of orchid species in the studied zone, reference to Piotrowska (1994). The sentence from line 278-279 ("Also, it has...") will be deleted - it has no substantive significance. The sentence from lines 279-282 ("It past...") has been moved to the discussion section. All terminological, linguistic and interpretation comments will be

included in the final version of the article in accordance with the reviewer's comments.

---

## Author Comment (AC11) · 15 Oct 2020

- climate indicator formulas will be incorporated into working methods - part of the text from the Conclusions chapter (In past elaborations, the indicatory species of Cephalantero rubra featured a larger reach in the area of Wolin National Park, e.g., in forest units 16 and 10 (WPN 2014 – Draft Plan of the Wolin National Park Protection for the years 2014 - 2033. The plan of flora and fungi protection. Internal materials, 19, 2014).

[Figure]

Currently, no specimens of Cephalantero rubra have been found on those sites, which is the confirmation for the decreasing reach of this species in Wolin National Park) will also be moved to the Discussion chapter to the line 271. - figures will be graphically corrected and statistical measures (R2, r, p and regression equation) will be added - photo from the research area will also be added - punctuation errors will be corrected and references to literature will be added

---

## Author Comment (AC12) · 15 Oct 2020

- in the Methods section, formulas for climate indicators will be added and threshold values for Fagus Silvatica will be described. De Martonne Aridity Index: $IA=P/(T+10)$ (De Martonne 1926), where P the amount of the annual precipitation, T average annual temperature. $IA<30$= silvosteppe, $30<IA<45$ climate favourable for the forest, with an optimal for beech in the range 35-40 (Satmari, 2010). De Martonne Aridity Index

[Figure]

- classification Tabari et al., 2014: IA<5 extremely arid 5<IA<10 arid 10<IA<20 semi-arid 20<IA<24 mediterranean 24<IA<28 semi-humid 28<IA<35 humid 35<IA<55 very humid 55<IA extremely humid. Ellenberg Quotient Index: EQ=Tw/Px1000 (Ellenberg 1988) where Tw represents the temperature of the warmest month of the year, P annual precipitations (Stojanovic et al., 2013). Ellenberg (1988) has set a threshold of beech favourability for EQ values lower than 30, and at EQ values that are higher than 40, the beech disappearance occurs. Forestry Aridity Index: FAI=100x(TVII-VIII/(PV-VII+PVII-VIII) where TVII-VIII is the average temperature of the months July and August, PV-VII represents the amount of precipitations during May-July and PVII-VIII is the amount of precipitations during July-August (Führer et al. 2011). Mayr Tetratherm Index: MT=(TV+TVI+TVII+TVIII)/4 where tV-tVIII represent the mean temperature for the May-August period. - A sentence will be completed in the summary (a few words in bracket in line 20-21): It has been established that in the 21st century, a relatively larger hazard to the functioning of the researched site are climate changes (ie mostly changes in thermal conditions and precipitation conditions) not the sea coast erosion.

---

## Author Comment (AC13) · 15 Oct 2020

- line 100 Added information on the statistical methods used, p value, regression equations, determination and correlation factors - line 164, 165, and 180 R2 value, regression equation, and statistical significance included in the figures - line 189 unified terminology as climatic indicators showing the impact of long-term weather conditions on the beech forest condition and development - in paragraph (line 219) information

on aeolian deposition will be added: The average annual rate of aeolian deposition on the cliff crown was almost 2 mm and the maximum (point) even 16 mm (Hojan M., 2009: Aeolian processes on the cliffs of Wolin Island. Quaestiones Geographicae 28/2: 39-46). The placed benchmarks showed an average aeolian deposition about 4-6 cm in the 2001 - 2020 period, with a maximum point increment of 10-12 cm. - The discussion will compare the rate of cliff erosion on Wolin Island to other research sections in the South Baltic zone (e.g. Florek W., Kaczmarzyk J., Majewski M., 2009: Intensity and character of cliff evolution near Ustka. Quaestiones Geographicae 28A/2: 27-38; ŁĂŹczyński L., 1999. Morpholitodynamics of the shoreface on the cliff coast at JastrzĂŹbia Góra. Peribalticum VII: 9-20; Uścinowicz G., Jurys L., Szarafin T., 2017. The development of unconsolidated sedimentary coastal cliffs (PobrzeÅije Kaszubskie, Northern Poland). Geological Quarterly 61(2): 491-501)
* * *

---

## Author Response (AR1)

[revised manuscript text omitted]

**2 Study Area and Methods**

**Z komentarzem [JT4]:** ANONYMOUS REFEREE (line 32): exchange word from valuable to all

**Z komentarzem [JT5]:** ANONYMOUS REFEREE (line 33): exchange word from stenotermal coastal to coastal thermophilous

**Z komentarzem [JT6]:** ANONYMOUS REFEREE (line 35): exchange word from complex to association

**Z komentarzem [JT7]:** ANONYMOUS REFEREE (line 36): exchange word from phytocoenosis to plant community

**Z komentarzem [JT8]:** ANONYMOUS REFEREE (line 36): exchange word from site to habitat

**Sformatowano:** Czcionka: Kursywa

**Z komentarzem [JT9]:** ANONYMOUS REFEREE (line 76-80): sentence was moved as recommended. Soil genetic levels added

**Z komentarzem [JT10]:** MR. KOZŁOWSKI added values for aeolian deposition

**Sformatowano:** Czcionka: Kursywa

**Sformatowano:** Indeks górny

**Sformatowano:** Indeks górny

**Z komentarzem [JT11]:** ANONYMOUS REFEREE (line 41). The text was supplemented in accordance with the comments of the reviewer

**Sformatowano:** Czcionka: Kursywa

**Sformatowano:** Czcionka: Kursywa

**Z komentarzem [JT12]:** ANONYMOUS REFEREE (line 42): exchange word from phytocoenotic to phytosociological

**Z komentarzem [JT13]:** ANONYMOUS REFEREE (line 43): exchange word from reach to plant richness

**Z komentarzem [JT14]:** ANONYMOUS REFEREE (line 44): exchange word from phytocoenosis to association

**Z komentarzem [JT15]:** ANONYMOUS REFEREE (line 46-56): Historical aspects of Cr-F were removed as suggested

~~The known history of *Cr-F* growth on Wolin Island dates back to the end of the 18th century, when natural beech and oak sites had been cut down (with the exception of a small number of the so-called 'parents of family') and pine monoculture was introduced. Such an unfavourable action led to unification of tree sites, acidification and impoverishment of the soil (Piotrowska, 1993), as well as decay of the primary *Cr-F* site. Then, as an outcome of aeolian supply of mineral matter from the cliff slope onto cliff top, a soil started to develop in the form of cliff naspa (Prusinkiewicz, 1971). Accumulation of naspa mechanically destroyed pine forest ground cover and created conditions for re-settlement of species with more extensive trophic demands (including neutrophils) and favoured growth of beech share within pine sites. Current age of beeches in *Cr-F* sites is 150–185 years (Piotrowska, 1993). Thus, the oldest of the currently existing beeches grew in the beginning of naspa accumulation period. Natural expansion of the beech advanced and catered for re-establishment of *Cr-F* (Piotrowska 1993), which occurs until the present day. It should be stressed that paleogeographical sediments record of the 'primary' *Cr-F* site from before the 18th century is not available, as this part of the cliff coast was subject to coastal erosion.~~

The section of cliff coast, in which *Cr-F* occurs, was developed as a result of undercutting Wolin end moraine by the transgressing Baltic Sea. Ultimately, orchid beech wood sites have been developed on hinterland of moraine cliffs. Moraine cliffs at *Cr-F* sites are characterised by high morphological (height of 20–95 m, dominant NW exposition, inclinations op to 1° on cliff top, and up to 88° on clayey slopes) and lithological (sandy sections, clayey or mixed — sandy and clayey) differentiation. The analysed section of cliff coast with the length of merely 3 km features various morphodynamic  states (erosion or stagnation). The researched site type is rich in species characteristic for, both, forest and non-forest phytocoenoses. Forest species,  characteristic for *Fagetalia* and *Querco-Fagetea* as well as meadow species with *Molinio-Arrhenatheretea* occur in large numbers (Piotrowska, 1993). The high flow of light to the ground from the sea direction favours the occurrence on the top cliff of many heliophilous species, characteristic for meadows and psammophilous short-grass swards. Gramineous species prevail in the herb layer, among others: *Brachypodium sylvatica*, *Dactylis glomerata*, *Poa Nemoralis*. The most valuable are orchid species, *Cephalanthera damasonium*, *Cephalanthera rubrae*,  *Epipaptis atrorubens*, which prefer fertile soils with reaction close to neutral (Piotrowska, 2003). There are, however, no of the numerous species characteristic for *Fagetalia*  *sylvaticae* order ( *Actaea spicata*, *Daphne mezereum*, *Lathyrus vernus*, *Mercurialis perennis*) and *Querco-Fagetea* class (*Aegopodium podagraria*, *Campanula trachelium*, *Corylus avellana*) that feature considerable share in all other *Cephalanthero-Fagenion* forests, which evidences the distinction and uniqueness of the *Cr-F* association  (Matuszkiewicz, 2001). The source of Latin names of plant species and plant communities are the publications Jackowiak et al. (2007) and Matuszkiewicz (2020).

The current reach and floristic composition of *Cr-F* has been determined on the basis of a few phytosociological mapping conducted on 6 study sites over 2018 and 2019 vegetative seasons. All in all, 10 detailed phytosociological images were taken with the use of Braun–Blanquet method, and *Cr-F* habitats  reach chart on Wolin island was drafted (Fig. 1).

**Sformatowano:** Wcięcie: Pierwszy wiersz: 1 cm

**Z komentarzem [JT16]:** ANONYMOUS REFEREE (line 63): exchange word from typical to characteristic

**Z komentarzem [JT17]:** ANONYMOUS REFEREE (line 64-65): sentence changed as suggested

**Z komentarzem [JT18]:** ANONYMOUS REFEREE (line 65): exchange word from ground cover to herb layer

**Z komentarzem [JT19]:** ANONYMOUS REFEREE (line 66): alphabetical order

**Z komentarzem [JT20]:** ANONYMOUS REFEREE (line 70): exchange word from orchid beech woods to Cephalanthero-Fagenion forests

**Sformatowano:** Czcionka: Kursywa

**Z komentarzem [JT21]:** ANONYMOUS REFEREE (line 71): exchange word from complex to association

**Z komentarzem [JT22]:** ANONYMOUS REFEREE (line 71-72): Sentence removed. It was already on lines 35-39

**Z komentarzem [JT23]:** ANONYMOUS REFEREE (line 45): Text added as suggested. Source of Latin names

**Z komentarzem [JT24]:** ANONYMOUS REFEREE (line 75): exchange word from site to habitats

An assumption was adopted that *Cr-F* site reach is determined by soil conditions. , The cliff naspa determines the occurrence of *Cephalanthera rubra* and *Epipactis artorubens*, which are species regionally characteristic of *Cephalanthero rubrae-Fagetum*. as the cliff naspa conditions occurrence of some of *Orchidaceae* family species. Naspa's accumulation level consists in interbeddings of fine-grain sand and dust drifted by wind from eroded cliff slopes, and rich in humus, dark-grey organic accumulation laminas (mainly leaves of *Fagus Silvatica*). The cliff naspa is a soil with reaction close to neutral, rich in calcium carbonate and characterised by high porosity and efficient humification of organic remains. That is why naspa is a fertile soil (Prusinkiewicz, 1971). The site's reach limits are indicated on the basis of occurrence of *Cephalanthera rubra*, that is an indicatory species for *Cr-F* complex.

**Sformatowano:** Czcionka: Kursywa

**Sformatowano:** Czcionka: Kursywa

**Sformatowano:** Czcionka: Kursywa

**Z komentarzem [JT25]:** ANONYMOUS REFEREE (line 76-77): sentence changed as suggested

**Z komentarzem [JT26]:** ANONYMOUS REFEREE (line 81): the sentence was shortened

[Figure]

[Figure]

**Figure 1.** *Cr-F*  habitats, localisation of phytosociological mapping on Wolin Island.

**Z komentarzem [JT27]:** exchange word from site to habitats

[Figure]

**Figure 2.** *Cr-F* habitat II on Wolin Island.

Detailed recognition of hydrometeorological conditions and the recession rate of the cliff top are vastly important for the functioning of *Cr-F* habitats. Thermal and precipitation conditions determine, e.g. on water and heat resources and duration of vegetative season. On the other hand, extreme storm surges may generate intensive cliff erosion and consequently reduce the spatial extent of coastal plant communities. Therefore, unfavorable hydrometeorological conditions may limit the development of the *Cr-F* habitats. For the purpose of defining long-term trend for thermal and precipitation conditions and sea level, daily hydrometeorological data in the period of 1960–2019, collected in measurement station in Świnoujście, were used. The data were provided by the Polish Institute of Meteorology and Water Management. The meteorological and mareographical station in  Świnoujście is located 15 km from the research area and provides homogeneous  and complete series of actual data.

In the elaboration, a number of especially useful climatic indicators were calculated and their values compared with threshold values adequate for *Fagus*  *sylvatica* given by Budeanu et al. (2016):

- De Martonne  Aridity Index IA=P/(T+10),  (where *P* is the amount of the annual precipitation, *T* is the average annual temperature (De Martonne, 1926); with optimal thresholds for beech wood in the range of 35–40 (Satmari, 2010); De Martonne Aridity Index - classification by Tabari et al., (2014): IA<5 extremely arid, 5<IA<10 arid, 10<IA<20 semi-arid, 20<IA<24 mediterranean, 24<IA<28 semi-humid, 28<IA<35 humid, 35<IA<55 very humid, 55<IA extremely humid.
- Ellenberg Quotient EQ=Tw/Px1000, where *Tw* is the temperature of the warmest month of the year, *P* is the annual precipitations (Ellenberg, 1988); with optimal threshold beneficial for beech growth of below 30 and its recession threshold of above 40 (Stojanovic et al., 2013),
- Forestry Aridity Index FAI=100x($T_{VII-VIII}$/($P_{V-VII}$+$P_{VII-VIII}$), where $T_{VII-VIII}$ is the average temperature of the months July and August, $P_{V-VII}$ is the amount of precipitations during May-July and $P_{VII-VIII}$ is the amount of precipitations during July-August; with climatic conditions favouring beeches of below 4.75 (Führer et al., 2011),

**Sformatowano:** Wyrównany do środka

**Z komentarzem [JT28]:** MR. KOLANDER, MRS. KIJOWSKA an overview photo of the habitat has been added

**Sformatowano:** Wcięcie: Pierwszy wiersz: 1,25 cm

**Sformatowano:** Czcionka: Kursywa

**Z komentarzem [JT29]:** ANONYMOUS REFEREE (line 84-86): the text has been completed as suggested

**Z komentarzem [JT30]:** ANONYMOUS REFEREE (line 88): text corrected as suggested

**Sformatowano:** Odstęp Przed: 0 pkt

**Z komentarzem [JT31]:** ANONYMOUS REFEREE (line 90): corrected citation

**Sformatowano:** Akapit z listą, Wcięcie: Z lewej: 0 cm, Wysunięcie: 0,25 cm, Odstęp Przed: 0 pkt, Punktowane + Poziom: 1 + Wyrównanie: 0,63 cm + Wcięcie: 1,27 cm

**Sformatowano:** Czcionka: Kursywa

**Sformatowano:** Czcionka: Kursywa

**Sformatowano:** Czcionka: Kursywa

**Sformatowano:** Indeks dolny

**Sformatowano:** Indeks dolny

**Sformatowano:** Indeks dolny

**Sformatowano:** Czcionka: Kursywa

**Sformatowano:** Czcionka: Kursywa, Indeks dolny

**Sformatowano:** Czcionka: Kursywa

**Sformatowano:** Czcionka: Kursywa, Indeks dolny

**Sformatowano:** Czcionka: Kursywa

**Sformatowano:** Czcionka: Kursywa, Indeks dolny

**Sformatowano:** Indeks dolny

[revised manuscript text omitted]

**Sformatowano:** Czcionka: Kursywa

**Z komentarzem [JT37]:** MR. WELSH: Raw data source added as suggested

**Z komentarzem [JT38]:** MR. KOZŁOWSKI, MRS. KIJOWSKA
R2 value, regression equation,
correlation index and p-value included in the figures

**Sformatowano:** Czcionka: Kursywa

**Sformatowano:** Czcionka: Kursywa

**Z komentarzem [JT39]:** ANONYMOUS REFEREE (line 189): exchange word from ground cover to herb layer

[Figure]

**Figure 5.** Long-term trends in climatic indicators: De Martonne Aridity Index (AI), Ellenberg Quotient (EQ), Forestry Aridity Index (FAI), Mayr Tetratherm Index (MT), (Świnoujście, 1960–2019). *(Own study based on raw data from the Institute of Meteorology and Water Management in Warsaw).*

**3.2 Cliff Coast Morphodynamics Hazard**

The mean annual rate of cliff top recession in 1984–2019 at *Cr-F* habitats II, III and V amounted to 0.24 m $yr^{-1}$. The lowest mean annual value of cliff recession was measured for site V (0.12 m $yr^{-1}$), where the cliff is built mainly of clayey sediments. The clayey sediments are characterised by relatively high resistance to degradation processes and the reaction time of cliff top to abrasion undercuttings is extended. A large number of storms is needed for the damages to reach the cliff top. On the other hand, the highest rate of cliff erosion has been established for site III (0.31 m $yr^{-1}$), where the cliff is built mainly of sandy material that is non-resistant to erosion. Sandy sediments are characterised by very low cohesion and are subject of rapid degradation. During stormy swellings, the sandy cliffs are undercut in a short time, which favours initiation of aeolian processes (deflation) and mass movements (sheddings, slidings). The processes cause the sediments to move across the entire slope profile, and thus the reaction of cliff top to abrasion undercutting is relatively short. An increased erosion dynamics has been observed also in site II (0.27 m $yr^{-1}$), on the cliff built of, both, clayey and sandy sediments. Its characteristic feature is the occurrence of underground water effluences, and high humidity of clayey sediments increases the susceptibility to landslide processes. The efficiency of the cliff springs is rather small <1 $dm^3$ $min^{-1}$. Landslide processes generate the highest cliff's transformations, contributing to movements of its top and cause reduction of *Cr-F* site area. In total, over the last 35 years, the researched cliffs recessed by an average of 7.32 m. The rate of recession of cliff top was spatially varied. The largest local and pinpoint movements were measured in the western part of site II (28.44 m) (Fig. 6). In this location, owing to high activity of landslide processes, the cliff top recessed with a high rate of 0.81 m $yr^{-1}$. In turn, the smallest local movements of cliff top were noted for eastern and western part of site V (0.30–0.42 m). In these locations, a very small rate of cliff top recession was connected with high resistance of clayey sediments to erosion processes and amounted to merely 0.01 m $yr^{-1}$.

**Sformatowano:** Czcionka: Kursywa

**Z komentarzem [JT40]:** MR. WELSH: Raw data source added as suggested

**Z komentarzem [JT41]:** MR. KOZŁOWSKI, MRS. KIJOWSKA
R2 value, regression equation, correlation index and p-value included in the figures

**Sformatowano:** Czcionka: Kursywa

**Sformatowano:** Czcionka: Kursywa

**Z komentarzem [JT42]:** ANONYMOUS REFEREE (line 207): the text has been completed as suggested

**Sformatowano:** Indeks górny

**Sformatowano:** Indeks górny

[year]

-1
-2
-3
-4
-5
[m]
A
Site II
Site III
Site V

[year]

-1
-2
-3
-4
-5
[m]
A
Habitat II
Habitat III
Habitat V

[Figure]

[Figure]

**Figure 6.** Location changes of cliff top at sites II, III and V of *Cr-F* in the period of 1985–2019: A – annual mean at sites, B – total multiannual in measurement points at sites. *(Own study based on own measurements and raw data from Kostrzewski et al. 2015, Winowski et al. 2019).*

A relatively lower sections of cliff coast (<30 m a.s.l.), which are primarily built of non-resistant to erosion sandy formations, do not favour the occurrence of the *Cr-F* phytocoenoses. In these sections of cliff coast, the deposition of sediments containing the calcium carbonate required by the orchid beech wood is relatively small (sandy sediments contain 4-5 times less calcium carbonate 2% than clay sediments) and an increased erosion (sandy sediments are much less resistant to erosion than clay sediments) of the coast results additionally in the reduction of habitat's area. A different situation is with the high cliff, with considerable share of clayey sediments. When aeolian processes occur, the dusty material, originating mainly in the clayey slope, rich in calcium carbonate, is accumulated on the cliff top and in cliff's hinterland, causing soil

**Sformatowano:** Czcionka: Kursywa

**Z komentarzem [JT43]:** MR. WELSH: Raw data source added as suggested

**Sformatowano:** Czcionka: Kursywa

**Sformatowano:** Czcionka: Kursywa

**Z komentarzem [JT44]:** ANONYMOUS REFEREE (line 219): the text has been supplemented as suggested

**Z komentarzem [JT45]:** ANONYMOUS REFEREE (line 220): exchange words from orchid beech wood to Cr-F phytocoenoses

**Sformatowano:** Czcionka: Kursywa

**Z komentarzem [JT46]:** ANONYMOUS REFEREE (line 221): clarification added

deacidification. This is the condition that particularly favours the development of *Cr-F* habitats (e.g., habitat V). Limited occurrence of the orchid beech wood or its lack stems also from development cycles of the cliff coast. For the sandy and dusty material — that is the components of the cliff naspa — to be supplied, a morphogenetic activity at the cliff's slope is required. Only then material deflation from the cliff's slope and its subsequent aeolian deposition in the cliff's hinterland is possible. Thus, the aeolian deposition is indispensable for the formation and development of the cliff naspa for inland. When the cliff coast, over an extended period of time, is not subject to processes of maritime abrasion and slope erosion, then its slope is covered with vegetation under of biocenotic succession. The vegetation considerably hinders, and even renders impossible the supply of aeolian matter, and, in consequence, the formation of cliff naspa, which in a longer perspective spurs the decay of the *Cr-F* phytocoenoses (e.g., habitat I). That is the occurrence of the active morphogenetic processes of small intensity is desirable (e.g., at habitat V, mean annual rate of cliff top recession in the last 35 years amounted to 'as little as' 0.12 m $yr^{-1}$). The dynamics of coast recession may not, however, be too intensive, and exceed the natural expansion of the cliff naspa and *Cephalanthero rubrae-Fagetum* habitat for inland direction. Then, the decrease in habitat area is spurred (e.g., on habitat III, mean annual rate of cliff top recession in the last 35 years has been considerable and amounted to 0.32 m $yr^{-1}$). Therefore, the optimal morpholitodynamic conditions for the growth of *Cr-F* are found mainly on habitat V. Similar conditions are on habitats II and IV. On the remaining habitats of the *Cr-F* phytocoenoses, the morpholitodynamic conditions are rather unfavourable - too much (habitat III) or too little (habitat I) cliff erosion.

**3. 3 Reach and Floristic Composition of *Cr-F**

Currently, *Cr-F* grows along the northern cliffed coast of Wolin island, between  Biała  Góra and Grodno, in 6 isolated sites with total area of merely 7.3 ha. The researched phytocoenosis occurs over a short, 3 km section of the coast, in the form of narrow belt of approximately 100 m for inland, between cliff's edge and a complex of  lowland acidophilous beech forest, *Luzulo pilosae-Fagetum*.

The floral richness of *Cr-F* association  consists in 113 species of vascular plants. They represent 2 divisions —  *Pteridophyta* and *Spermatophyta*. In  *Pteridophyta*  4 species have been recorded: *Dryopteris carthusiana*, *Dryopteris filix-mas*, *Polypodium vulgare* and *Pteridium aquilinum* . And, in *Spermatophyta*  3 classes have been confirmed: *Pinopsida* (2 species: *Juniperus communis* and *Pinus sylvestris*), *Magnoliopsida* (23 orders, 29 families and 82 species) and *Liliopsida* (respectively 3, 6 and 27). The richest in species have been the families of: *Poaceae* (14 species), *Asteraceae* (13 ), *Fabaceae* (11 ) and *Rosaceae* (6 ). *Orchidaceae*  has been represented by 7 species: *Cephalanthera damasonium*, *Cephalanthera rubra*, *Corallorhiza trifida*,  *Epipactis atrorubens*, *Epipactis hellaborine*, *Neottia nidus-avis*,  *Platanthera bifolia*. The researched site is an example of a coexistence between forest species of fertile and acidic beech woods, acidophilic oak woods and forests, and species of meadows and psammophilous swards.

**Z komentarzem [JT47]:** ANONYMOUS REFEREE (line 229): exchange words from movement to development

**Z komentarzem [JT48]:** ANONYMOUS REFEREE (line 230): exchange words as suggested

**Sformatowano:** Czcionka: Kursywa

**Sformatowano:** Czcionka: Kursywa

**Z komentarzem [JT49]:** ANONYMOUS REFEREE (line 239): the text has been supplemented as suggested

**Z komentarzem [JT50]:** ANONYMOUS REFEREE (line 105): exchange words from acidic fertile lowland beech wood to lowland acidophilous beech forest

**Z komentarzem [JT51]:** ANONYMOUS REFEREE (line 107): exchange word from complex to association

**Sformatowano:** Czcionka: Nie Kursywa

**Z komentarzem [JT52]:** ANONYMOUS REFEREE (line 108): the text has been changed as suggested

[revised manuscript text omitted]

**Sformatowano:** Indeks górny

**Sformatowano:** Czcionka: Kursywa

**Sformatowano:** Nie Wyróżnienie

**Sformatowano:** Czcionka: Kursywa

**Z komentarzem [JT68]:** ANONYMOUS REFEREE (line 131-135): added quantitative description

**Sformatowano:** Czcionka: Kursywa

**Sformatowano:** Nie Wyróżnienie

**Sformatowano:** Indeks górny

**Z komentarzem [JT69]:** ANONYMOUS REFEREE (line 128-129): more information have been added

**Sformatowano:** Nie Wyróżnienie

**Sformatowano:** Czcionka: Kursywa

**Z komentarzem [JT70]:** ANONYMOUS REFEREE (line 136-142): more information have been added

**Sformatowano:** Czcionka: Kursywa

**Sformatowano:** Nie Wyróżnienie

**Sformatowano:** Czcionka: Kursywa

**Sformatowano:** Czcionka: Kursywa

**Z komentarzem [JT71]:** ANONYMOUS REFEREE (line 143-144): more information have been added

**Sformatowano:** Czcionka: Kursywa

**Sformatowano:** Indeks górny

[revised manuscript text omitted]

**Sformatowano:** Polski

**Sformatowano:** Polski

**Sformatowano:** Polski

**Sformatowano:** Polski

**Sformatowano:** Domyślna czcionka akapitu, Angielski (Zjednoczone Królestwo)

**RESPONSES TO REVIEWS AND COMMENTS**

**All comments of the reviewer will be included in the article:**

**(line 1)**
ANONYMOUS REFEREE (line 1): I propose to shorten the title: Influence of Hydrometeorological Hazards and Sea Coast Morphodynamics onto Development of the Cephalanthero rubrae-Fagetum (Wolin Island, the Southern Baltic Sea)
ANSWER JT (line 1): "The title was shortened: Influence of Hydrometeorological Hazards and Sea Coast Morphodynamics onto Development of the *Cephalanthero rubrae-Fagetum* (Wolin Island, the Southern Baltic Sea)"

**(line 12, 13, 18)**
ANONYMOUS REFEREE (line 13, 14, 19): remove the space
ANSWER JT (line 12, 13, 18): spaces have been removed

**(line 18-20)**
MR. WOLSKI (3) (lines 19-21): Please add a few words in bracket in the sentence in Abstract: (lines 19-21): "It has been established that in the 21st century, a relatively larger hazard to the functioning of the researched site are climate changes (ie mostly changes in thermal conditions and precipitation conditions) not the sea coast erosion" This will be clear to the reader.
ANSWER JT (line 18-20): the sentence was completed as suggested: "It has been established that in the 21st century, a relatively larger hazard to the functioning of the researched site are climate changes (i.e. mostly changes in thermal and precipitation conditions) not the sea coast erosion."

**(line 31)**
ANONYMOUS REFEREE (line 32): all, not only valuable
ANSWER JT (line 31): exchange word from valuable to "all"

**(line 32)**
ANONYMOUS REFEREE (line 33): coastal thermophilous
ANSWER JT (line 32): exchange word from stenotermal coastal to "coastal thermophilous"

**(line 34)**
ANONYMOUS REFEREE (line 35): association
ANSWER JT (line 34): exchange word from complex to "association"

**(line 35)**
ANONYMOUS REFEREE (line 36): plant community
ANSWER JT (line 35): exchange word from phytocoenosis to "plant community"

**(line 35)**
ANONYMOUS REFEREE (line 36): habitat

ANSWER JT (line 35): exchange word from site to "habitat"

**(line 37-43)**

ANONYMOUS REFEREE (line 76-80): the cliff naspa determines the occurrence of Cephalanthera rubra and Epipactis artorubens, which are species regionally characteristic of Cephalanthero rubrae-Fagetum. transfer and combine with information onto lines 35-39. Give there the full description of the naspa according to Prusinkiewicz 1971; change Silvatica into sylvatica

ANSWER JT (line 37-43): sentence was moved as recommended. Soil genetic levels added. "Naspa's accumulation level consists in interbeddings of fine-grain sand and dust drifted by wind from eroded cliff slopes, and rich in humus, dark-grey organic accumulation laminas (mainly leaves of *Fagus sylvatica*). The cliff naspa is a soil with reaction close to neutral, rich in calcium carbonate and characterised by high porosity and efficient humification of organic remains. That is why naspa is a fertile soil. Naspa is deposited on the fossil podzolic soil. Naspa has the following sequence of soil levels: A0 litter level; A1I accumulation level of sand and organic matter layers; A1 (fos) accumulation level of fossil podzolic soil; A2 (fos) eluvial level of fossil podsolic soil; B (fos) iluvial level of fossil podzolic soil; C (fos) parent rock of fossil podzolic soil (Prusinkiewicz, 1971)." Exchange word from silvatica to "sylvatica"

**(line 45-47)**

MR. KOZŁOWSKI: The Authors wrote: "In these sections of cliff coast, the deposition of sediments containing the calcium carbonate required by the orchid beech wood is relatively small. . .". Did the Authors examine the amount of the deposition size?

ANSWER JT (line 37-43): added values for aeolian deposition "The average rate of aeolian deposition in the *Cr-F* habitat was 3-5 mm $y^{-1}$, and the maximum point value was 8-10 mm $y^{-1}$ (2000-2019)."

**(line 48-50)**

ANONYMOUS REFEREE (line 41). The authors should include the phytosociological characteristics of Cephalanthero rubrae-Fagetum association here (according to the syntaxonomy of Matuszkiewicz 2012), because the text repeatedly refers to typical Cr-F patches or typical species (lines: 124, 136, 140, 143, 145). Plant species regionally characteristic for Cr-F and characteristic for Cephalanthero-Fagenion should be given. Only Cephalanthera rubra and Epipactis atrorubens testify to the presence of well-developed patches of the association. If only Cephalanthera damasonium and Epipactis helleborine are present, phytocoenosis can only be included in the Cephalanthero-Fagenion compound and it is not a typical Cr-F

ANSWER JT (line 48-50): The text was supplemented in accordance with the comments of the reviewer "*Cephalanthera rubra* and *Epipactis atrorubens* are indicator species for *Cr-F* (Matuszkiewicz, 2020). Both species found in the 6 studied *Cr-F* habitats, but *Cephalanthera rubra* was the dominant one. Non-indicator species, e.g. *Cephalanthera damasonium* and *Epipactis helleborine*, have been found in *Cr-F* habitats too."

**(line 48)**

ANONYMOUS REFEREE (line 42): phytosociological

ANSWER JT (line 48): exchange word from phytocoenotic to “phytosociological”

**(line 53)**
ANONYMOUS REFEREE (line 43): plant richness
ANSWER JT (line 53): exchange word from reach to “plant richness”

**(line 54)**
ANONYMOUS REFEREE (line 44): association
ANSWER JT (line 54): exchange word from phytocoenosis to “association”

**(line 56)**
ANONYMOUS REFEREE (line 46-56): Since the entire habitat occupied by the pine monoculture does not exist today, as it has been eroded (line 55 and 56), this description is absolutely unnecessary. It should be removed. The described story does not concern the places where the phytocoenoses studied by the authors occur today. Therefore, we cannot talk about the return of the habitat, but about the development of new habitats - the way of transforming those that found themselves at the edge of the cliff.
ANSWER JT (line 56): Historical aspects of Cr-F were removed as suggested

**(line 60)**
ANONYMOUS REFEREE (line 61): The concept of morphodynamic functions is not used in the world geomorphological literature. Replace with another, e.g. morphodynamic states or morphodynamic processes
ANSWER JT (line 60): exchange word from functions to “states”

**(line 62)**
ANONYMOUS REFEREE (line 63): characteristic
ANSWER JT (line 62): exchange word from typical to “characteristic”

**(line 63-64)**
ANONYMOUS REFEREE (line 64-65): If increased, relative to what?: Light characteristic for meadows and psammophilous short-grass swards.
ANSWER JT (line 63-64): sentence changed as suggested ). “The high flow of light to the ground from the sea direction favours the occurrence on the top cliff of many heliophilous species, characteristic for meadows and psammophilous short-grass swards.”

**(line 65)**
ANONYMOUS REFEREE (line 65): herb layer
ANSWER JT (line 65): exchange word from ground cover to “herb layer”

**(line 65-66)**
ANONYMOUS REFEREE (line 66): why not in alphabetical order, Dactylis glomerata and Poa nemoralis.

ANSWER JT (line 65-66): alphabetical order of species

**(line 68)**
ANONYMOUS REFEREE (line 68): sylvaticae, Actaea
ANSWER JT (line 68): typing errors were corrected

**(line 70)**
ANONYMOUS REFEREE (line 70): Cephalanthero-Fagenion forests
ANSWER JT (line 70): exchange word from orchid beech woods to "*Cephalanthero-Fagenion* forests"

**(line 70)**
ANONYMOUS REFEREE (line 71): association
ANSWER JT (line 70): exchange word from complex to "association"

**(line 71-72)**
ANONYMOUS REFEREE (line 71-72): It was already on lines 35-39. Remove
ANSWER JT (line 71-72): Sentence removed.

**(line 71-72)**
ANONYMOUS REFEREE (line 45): in this chapter to give according to whom the Latin names of plant species and phytosociological units were given
ANSWER JT (line 71-72): Text added as suggested. Source of Latin names "The source of Latin names of plant species and plant communities are the publications Jackowiak et al. (2007) and Matuszkiewicz (2020)."

**(line 75)**
ANONYMOUS REFEREE (line 75): habitats
ANSWER JT (line 75): exchange word from site to "habitats"

**(line 76-77)**
ANONYMOUS REFEREE (line 76-77): "The cliff naspa determines the occurrence of Cephalanthera rubra and Epipactis artorubens, which are species regionally characteristic of Cephalanthero rubrae-Fagetum."
ANSWER JT (line 76-77): sentence changed as suggested

**(line 78)**
ANONYMOUS REFEREE (line 81): "…*rubra*."
ANSWER JT (line 78): The sentence was shortened as suggested

**(line 80)**:
JT: Figure 1. exchange word from site to "habitats"

**(line 81-82)**

MR. KOLANDER, MRS. KIJOWSKA: An overview photo of the habitat has been added
ANSWER JT (line 81-82): Photo of habitat added as suggested

**(line 84-87)**
ANONYMOUS REFEREE (line 84-86): justify why, what hydrometeorological parameter?
ANSWER JT (line 84-87): the text has been completed as suggested "Thermal and precipitation conditions determine, e.g. on water and heat resources and duration of vegetative season. On the other hand, extreme storm surges may generate intensive cliff erosion and consequently reduce the spatial extent of coastal plant communities. Therefore, unfavorable hydrometeorological conditions may limit the development of the *Cr-F* habitats."

**(line 88)**
ANONYMOUS REFEREE (line 87): Świnoujście
ANSWER JT (line 88): exchange word from Swinoujscie to "Świnoujście"

**(line 89-91)**
ANONYMOUS REFEREE (line 88): the Institute's data are always reliable, what does it mean?
ANSWER JT (line 89-91): text corrected as suggested ". The meteorological and mareographical station in Świnoujście is located 15 km from the research area and provides homogeneous and complete series of actual data."

**(line 93)**
ANONYMOUS REFEREE (line 89): sylvatica
ANSWER JT (line 93): exchange word from silvatica to "sylvatica"

**(line 93)**
ANONYMOUS REFEREE (line 90): given by Budeanu et al. (2016):
ANSWER JT (line 93): sentence corrected "In the elaboration, a number of especially useful climatic indicators were calculated and their values compared with threshold values adequate for *Fagus sylvatica* given by Budeanu et al. (2016):"

**(line 94-105)**
MR. WOLSKI (1): "The authors of the study identified interesting climatic indicators (AI, EQ, FAI, MT).
However, they were not well described. Please complete the formulas of these indicators. Please write how the value of a particular indicator influences the development (growth) of Fagus Silvatica"
ANSWER JT (line 94-105): formulas for climate indicators have been added

"-De Martonne Aridity Index IA=P/(T+10), where P is the amount of the annual precipitation, T is the average annual temperature (De Martonne, 1926); with optimal thresholds for beech wood in the range of 35–40 (Satmari, 2010); De Martonne Aridity Index - classification by Tabari et al., (2014): IA<5

extremely arid, 5<IA<10 arid, 10<IA<20 semi-arid, 20<IA<24 mediterranean, 24<IA<28 semi-humid, 28<IA<35 humid, 35<IA<55 very humid, 55<IA extremely humid.

-Ellenberg Quotient EQ=Tw/Px1000, where Tw is the temperature of the warmest month of the year, P is the annual precipitations (Ellenberg, 1988); with optimal threshold beneficial for beech growth of below 30 and its recession threshold of above 40 (Stojanovic et al., 2013),

-Forestry Aridity Index FAI=100x(TVII-VIII/(PV-VII+PVII-VIII), where TVII-VIII is the average temperature of the months July and August, PV-VII is the amount of precipitations during May-July and PVII-VIII is the amount of precipitations during July-August; with climatic conditions favouring beeches of below 4.75 (Führer et al., 2011),

-Mayr Tetratherm: MT=(TV+TVI+TVII+TVIII)/4, where TV-TVIII represent the mean temperature for the May-August period (Mayr, 1909); with optimal thermal conditions for beech wood of 13–18°C (Satmari, 2010).

**(line 113)**
ANONYMOUS REFEREE (line 102): First, the abiotic conditions for Cr-F should be characterized, and then the floristic and phytosociological characteristics of Cr-F using the given characteristics of habitat conditions. Because the range and floristic composition of Cr-F depend on them. Use the past tense throughout the chapter. It is a description of some past condition that does not exist now.
ANSWER JT (line 113, 158, 205): The order of subsections was changed as recommended: 3.1 Hydrometeorological Conditions and Hazards; 3.2 Cliff Coast Morphodynamics Hazard; 3.3 Reach and Floristic Composition of Cr-F

**(line 114)**
ANONYMOUS REFEREE (line 155): period (1960-2019)
ANSWER JT (line 114): the text has been completed as suggested "In the researched 60-year period (1960-2019)…"

**(line 123, 124, 139, 153, 154)**
MR. KOZŁOWSKI, MRS. KIJOWSKA: Please add R2 value, equation and statistical significance.
ANSWER JT (line 123, 124, 139, 153, 154): R2 value, regression equation, correlation index and p-value included in the diagrams (Figure 3, 4, 5)

**(line 126-127, 140-141, 156-157)**
MR. WELSH: data source should be added

ANSWER JT (line 126-127, 140-141, 156-157): Raw data source added as suggested in n the title of the figures 3, 4 and 5 “(Own study based on raw data from the Institute of Meteorology and Water Management in Warsaw)”

**(line 150)**
ANONYMOUS REFEREE (line 189): degeneration
ANSWER JT (line 150): exchange word from degradation to “degeneration”

**(line 196)**
ANONYMOUS REFEREE (line 196): In this chapter, please include only the results of your research carried out with the methods described in chapter 2. The authors describe the effects of processes they have not studied. This should be in the discussion chapter
ANSWER JT: the above remark was included in the text correction

**(line 159-161)**
ANONYMOUS REFEREE (line 198-199): In lines 198-199 is: the cliff is built mainly of clayey sediments.
In lines 148-149 is: with balanced share of clayey and sandy sediments. What does it mean, in the geological term: balanced share of clayey and sandy sediments.
ANSWER JT (line 259-262): sentence (line 259-262, earlier 147-151) corrected: ”Habitat V is the best developed patch of Cr-F, with optimal habitat conditions: favourable morpholitodynamic conditions (abrasive coast but low rate of cliff's recession 0.12 m yr-1, higher share of clay sediments, rich in calcium carbonate 8-10%); favourable light conditions (relatively greater insolation of the forest floor); ground cover of orchid beech wood, developing for inland for a dozen or so meters in some points).”

**(line 170)**
ANONYMOUS REFEREE (line 207): Can something more be written about the nature of these outflows? how much, where, how much water
ANSWER JT (line 170): sentence was added: ” The efficiency of the cliff springs is rather small <1 $dm^3$ $min^{-1}$.”

**(line 180-181)**
MR. WELSH: data source should be added
ANSWER JT (line 180-181): Raw data source added as suggested in n the title of the figure 6 “(Own study based on own measurements and raw data from Kostrzewski et al. 2015, Winowski et al. 2019).

**(line 182)**
ANONYMOUS REFEREE (line 219): enter height
ANSWER JT (line 182): the text has been supplemented as suggested “..(<30 m a.s.l.)..”

**(line 183)**
ANONYMOUS REFEREE (line 220): Cr-F phytocoenoses.

ANSWER JT (line 183): exchange words from orchid beech wood to “*Cr-F* phytocoenoses”

**(line 183-186)**

ANONYMOUS REFEREE (line 221): How do you know that? how much? how much increased? in relation to what?

ANSWER JT (line 183-186): clarification added: “…. (sandy sediments contain 4-5 times less calcium carbonate 2% than clay sediments) ….. (sandy sediments are much less resistant to erosion than clay sediments) …..”

**(line 193)**

ANONYMOUS REFEREE (line 229): Naspa does not move. It develops in situ in a beech forest that will be about 100 meters wide along the edge of the cliff.

ANSWER JT (line 193): exchange words from movement to “development”

**(line 195)**

ANONYMOUS REFEREE (line 230): vegetation under of biocenotic succession.

ANSWER JT (line 230): exchange words as suggested: from permanent crust vegetation to “vegetation under of biocenotic succession”

**(line 203-204)**

ANONYMOUS REFEREE (line 239): why?

ANSWER JT(line 203-204): the text has been supplemented as suggested”… - too much (habitat III) or too little (habitat I) cliff erosion.”

**(line 206)**

ANONYMOUS REFEREE (line 103): Biała Góra

ANSWER JT (line 206): exchange words from Biala Gora to “Biała Góra”

**(line 208)**

ANONYMOUS REFEREE (line 105): lowland acidophilous beech forest

ANSWER JT (line 208): exchange words from acidic fertile lowland beech wood to “lowland acidophilous beech forest”

**(line 210)**

ANONYMOUS REFEREE (line 107): association

ANSWER JT (line 210): exchange words from complex to “association”

**(line 211)**

ANONYMOUS REFEREE (line 108): Pteridophyta

ANSWER JT: the text has been changed as suggested

**(line 211-219)**

ANONYMOUS REFEREE (line 107-117): alphabetically or justify why in that order, Spermatophyta, sylvestris, respectively 3, 6 and 27, species – remove, alphabetically or justify why in that order, meadows and psammophilous swards. There have observed species from syntaxa, Artemisietea
ANSWER JT (211-219): the text has been changed as suggested, e.g. alphabetical order of species

**(line 222)**
ANONYMOUS REFEREE (e.g. line 118, 121, 124, 125, 131, 136, 138, 143): How was concentration tested and what is the result of these studies? Density (the number of individuals per unit of area) and size (how many individuals) population are properties of each population. Density is the number of individuals per area unit. Report the recorded density values of each of the four listed species? Cephalanthera rubra and Epipactis atrorubens are particullary important. They are regional characteristic species for Cr-F association. Enter the latitude and longitude of the center point
ANSWER JT (line 222): table with localisation and plant indicators of Cr-F habitats added as suggested. The quantitative data from the table are included in the description of habitats (line 224-260).

**(line 225)**
ANONYMOUS REFEREE (line 119): how much? how do you know this? please document, How do you know it goes away? Maybe it is just developing?
ANSWER JT (line 225): the added text answers the questions: "Therefore, aeolian deposition on the cliff top is very limited and the Cr-F habitat decays." The habitat is disappearing, not developing. For many years there has been no possibility of the naspa and habitat development.

**(line 225-227)**
ANONYMOUS REFEREE (line 119-121): The soil profile must have a morphology appropriate to the naspa - layer of aeolian sediments, etc. Only the results should be included in this chapter. All hypotheses and assumptions should be found in the discussion of results chapter.
ANSWER JT (line 225-227): The sentence needed for the specific functioning of the habitat

**(line 224-260)**
ANONYMOUS REFEREE
(line 121): For each site, the number of Cephalanthera rubra individuals recorded should be reported or a bioindicator such as the population density indicator of this species should be given, i.e. the number of individuals per square meter, distinguishing between vegetative and generative. Only the results should be included in this chapter. All hypotheses and assumptions should be found in the discussion of results chapter.
(line 121): For each site, the number of Cephalanthera rubra individuals recorded should be reported or a bioindicator such as the population density indicator of this species should be given, i.e. the number of individuals per square meter, distinguishing between vegetative and generative. Only the results should be included in this chapter. All hypotheses and assumptions should be found in the discussion of results chapter.

ANSWER JT (line 224-260): habitats localisation and quantitative indicators have been added (see Table 1), e.g. percentage of coverage in the herb layer, Terminology awkwardness fixed, e.g. line 229-230 "…..*Luzula pilosa* and *Trientalis europaea* are the distinguishing species of the *Luzulo-Fagenion* beech forests." Alphabetical order of species added.

**(line 238)**

ANONYMOUS REFEREE (line 130): A large portion of the site is covered by beech brushwood,
130 which evidences an intensive renewal of forest - move the assessment to the discussion chapter
ANSWER JT (line 238): The sentence needed for the specific functioning of the habitat. Sentence completed:"A large portion (20%) of the site is covered by beech brushwood, which evidences an intensive renewal of forest".

**(line 255-256)**

ANONYMOUS REFEREE (line 143-144): on what basis this an assumption?, patch of Cr-F typicum, what does a smaller concentration mean? what was the density and size population of each recorded orchid species? what does less concentration mean? what was the density and population size of each registered orchid species? especially Cephalanthera rubra and Epipactis atrorubens, which are regionally characteristic for Cr-F association, on what basis this an assumption?
ANSWER JT (line 255-256): more information have been added: "This habitat may also be considered a patch of *Cr-F* typicum (Table 1), but a smaller concentration of *Cephalanthera rubra* (15 individuals per ha) has been confirmed there. The cliff is mostly clayey and low (25-30 m a.s.l.), thus the intensity of aeolian deposition is relatively smaller (2 mm y-1 in 2000-2019)."

**(line 261)**

ANONYMOUS REFEREE (line 147): Cr-F

ANSWER JT (line 261): exchange words from orchid beech wood to" *Cr-F*"

**(line 261-266)**

ANONYMOUS REFEREE (line 148-152): In lines 198-199 is: the cliff is built mainly of clayey sediments. What does it mean, in the geological term: balanced share of clayey and sandy sediments, how much? Fractions, how rich? provide values for these parameters, What light conditions were favorable for the development of Cr-F phytocoenoses? remove, pinetisation was not discussed, the ground cover does not move for inland! The site does not decay away. The habitat conditions and floristic composition of the vegetation occurring at this site are changing.
ANSWER JT (line 261-266): sentence was corrected as suggested: "The most valuable orchid beech woods habitats are II, V and VI. Habitat V is the best developed patch of *Cr-F*, with optimal habitat conditions: favourable morpholitodynamic conditions (abrasive coast but low rate of cliff's recession 0.12 m $yr^{-1}$, higher share of clay sediments, rich in calcium carbonate 8-10%); favourable light conditions (relatively greater insolation of the forest floor); ground cover of orchid beech wood, developing for inland for a dozen or so meters in some points). The relatively poorest condition was confirmed for habitat I, which does not develop due to unfavorable morpholithodynamic conditions (dead non-erosive cliff, stabilised with compact pine wood, no possibility of forming naspa)."

**(line 297)**

ANONYMOUS REFEREE (line 270): will not be suitable for the development of the Cr-F habitat.
ANSWER JT (line 297): sentence was corrected as suggested

**(line 298-305)**

MR. KOZŁOWSKI: These is no reference to the observed changes in the position of the cliff in the discussion. Please add discussion with other authors
ANSWER JT (line 298-305): sentence was added as suggested "In the analysed period (1985-2019), the average annual rate of the cliff crown retraction on the examined sections amounted to 12 up to 31 cm and it was much lower than the values estimated (80-100 cm) by the mid-twentieth century by Subotowicz (1982) and Kostrzewski (1984). Whereas, the maximum annual point retraction of the cliff crown was almost 10 m. The average annual retraction rate of the Wolin cliffs is approximately 2-4 times lower than other monitored cliff coasts, e.g. in the vicinity of Ustka, Jastrzębia Góra or Gdynia (e.g., Florek et al. 2009; Łęczyński 1999). Although the Wolin cliffs are much higher and are not subjected to any protective measures, the relatively lowest rate of their retraction results primarily from specific hydrogeological conditions. For example, contrary to the cliff coast in Jastrzębia Góra (Uścinowicz et al. 2017) on the island of Wolin, underground waters practically do not play any role in erosion processes and shore degradation."

**(line 306-314)**

ANONYMOUS REFEREE (line 274-282), MRS. KIJOWSKA: Species composition of association's phytocoenoses, neither in the results nor in the discussion was the floristic composition of the patches 50 years ago compared to the present ones; on what basis this conclusion, Orchidaceae, Who and when found these orchids? They are not characteristic of either Cephalanthero rubrae-Fagetum, Cephalanthero-Fagenion, Fagetalia, or Querco-Fagetea. They don't have to keep up with the cliff's retreat. On what basis is this conclusion? There was no data on the current state of the population or a comparison with the state 50 years ago in the results, Why Lonicera is important for Cr-F? This species has little diagnostic value for Cr-F because it is a species characteristic of Querco-Fagetea. Transfer to discussion, cite the authors of these studies. Were these sites in Cr-F? Cephalanthera, On what basis this conclusion? The authors did not analyze the past and present geographical range of Cephalanthera rubra species in the national park.
ANSWER JT (line 306-314): sentence moved from conclusion to discussion "Species composition of association's phytocoenoses has not changed extensively over the last half-century (Piotrowska, 1993; Prusinkiewicz, 1971), which confirms its relative stability; however, some *Orchidaceae* habitats of do not keep up with the rate of the cliff's recession or they do not develop due to many years of cliff erosive stagnation. No specimens of *Malaxis monophyllos* were confirmed, which was occurring at the cliff's edge tens of years ago (Piotrowska, 1993). A vast loss for the site is also the lack of current confirmation for the occurrence of *Listera ovata*. Also, it has been confirmed that the number of *Lonicera xylosteum* decreased — a species important for the orchid beech wood. In past elaborations, the indicatory species of *Cephalantero rubra* featured a larger reach in the area of Wolin National Park, e.g., in forest divisions of Międzyzdroje 16 and Wisełka 2. Currently, no specimens of *Cephalantero*

*rubra* have been found on those sites, which is the confirmation for the decreasing reach of this species in Wolin National Park.

**(line 314)**

ANONYMOUS REFEREE (line 272-273): remove, The authors provided no evidence of habitat defragmentation. That there was a lobe that was divided into several. Cr-F habitats developed in different places, in scattered sites.

ANSWER JT (line 314): sentence was corrected as suggested “This valuable site consists of 6 isolated,…..”

**(line 328)**

ANONYMOUS REFEREE (line 291): Write what's going on?

ANSWER JT (line 328): more information have been added: “…. uncertainty of precipitation efficiency and their time distribution….”

In addition: linguistic inaccuracies and punctuation errors have been corrected, supplementing the necessary literature